# Adaptive Caching for Faster Video Generation with Diffusion Transformers

## Abstract

Generating temporally-consistent high-fidelity videos can be computationally expensive, especially over longer temporal spans. More-recent Diffusion Transformers (DiTs)— despite making significant headway in this context— have only heightened such challenges as they rely on larger models and heavier attention mechanisms, resulting in slower inference speeds. In this paper, we introduce a *training-free* method to accelerate video DiTs, termed Adaptive Caching (*AdaCache*), which is motivated by the fact that *"not all videos are created equal"*: meaning, some videos require fewer denoising steps to attain a reasonable quality than others. Building on this, we not only cache computations through the diffusion process, but also devise a caching schedule tailored to each video generation, maximizing the quality-latency trade-off. We further introduce a Motion Regularization (*MoReg*) scheme to utilize video information within AdaCache, essentially controlling the compute allocation based on motion content. Altogether, our plug-and-play contributions grant significant inference speedups (*e.g.* up to **4.7×** on Open-Sora 720p - 2s video generation) without sacrificing the generation quality, across multiple video DiT baselines. Our code will be made publicly-available.

## 1 Introduction

Diffusion models (Ho et al., 2020; Song et al., 2020) have become the standard for generative modeling in recent years, arguably surpassing the quality of VAEs (Kingma, 2013; Rolfe, 2016), GANs (Karras et al., 2019; Goodfellow et al., 2020) and Auto-Regressive models (Chang et al., 2022; 2023). This observation holds in a wide-range of applications including image (Rombach et al., 2022; Saharia et al., 2022), video (Singer et al., 2022; Blattmann et al., 2023a), 3D (Poole et al., 2022; Liu et al., 2023a), and audio (Kong et al., 2020; Huang et al., 2023) generation, as well as image (Hertz et al., 2022; Avrahami et al., 2023) and video (Qi et al., 2023; Wu et al., 2023) editing. More recent Diffusion Transformers (DiTs) (Peebles & Xie, 2023; Ma et al., 2024a) show better promise in terms of scalability and generalization compared to prior UNet-based diffusion models (Rombach et al., 2022), revealing intriguing horizons in GenAI for the years to come.

Despite the state-of-the-art performance, DiTs can also be computationally expensive both in terms of memory and computational requirements. This becomes especially critical when applied with a large number of input tokens (*e.g.* high-resolution long video generation). For instance, the reason for models such as Sora (OpenAI, 2024) not being publicly-served is speculated to be the high resource demands and slower inference speeds (Liu et al., 2024). To tackle these challenges and reduce the footprint of diffusion models, various research directions have emerged such as latent diffusion (Rombach et al., 2022), step-distillation (Sauer et al., 2023; Yin et al., 2024), caching (Wimbauer et al., 2024; Ma et al., 2024c; Habibian et al., 2024), architecture-search (Zhao et al., 2023b; Li et al., 2024b), token reduction (Bolya & Hoffman, 2023; Li et al., 2024a) and region-based methods (Nitzan et al., 2024; Kahatapitiya et al., 2024). Fewer techniques transfer readily from UNet-based pipelines to DiTs, whereas others often require novel formulations. Hence, DiT acceleration has been under-explored as of yet.

Moreover, we note that *not all videos are created equal*. Some videos contain high-frequency textures and significant motion content, whereas others are much simpler (*e.g.* with homogeneous textures or static regions). Having a diffusion process tailored specifically for each video generation can be beneficial in terms of realizing the best quality-latency trade-off. This idea has been explored

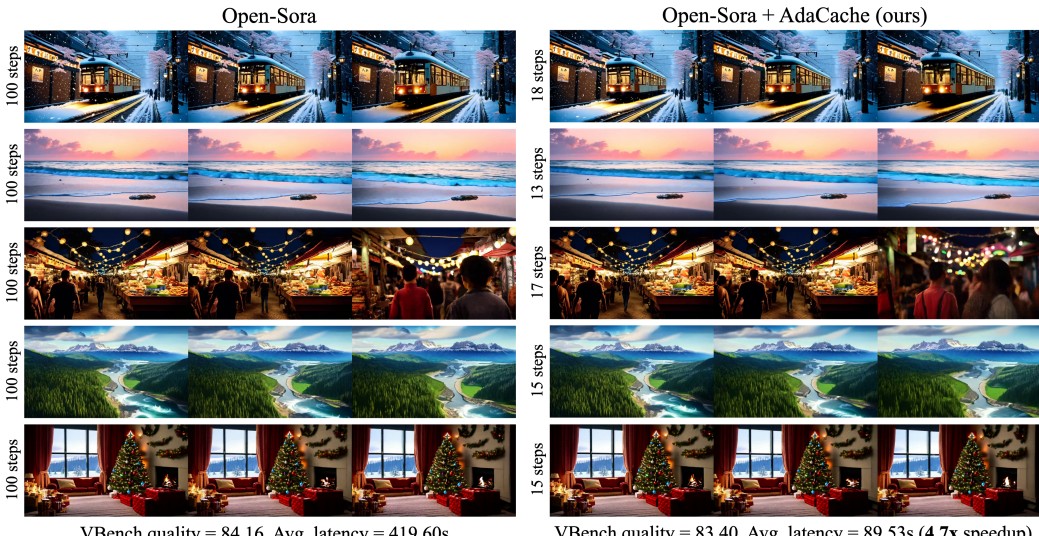

Figure 1: **Effectiveness of Adaptive Caching:** We show a qualitative comparison of AdaCache (right) applied on top of Open-Sora (Zheng et al., 2024) (left), a baseline video DiT. Here, we consider generating 720p - 2s video clips, and report VBench (Huang et al., 2024) quality and average latency on the standard benchmark prompts from Open-Sora gallery. AdaCache generates videos significantly faster (*i.e.*, 4.7× speedup) with a comparable quality. Also, the number of computed steps varies for each video. Best-viewed with zoom-in. Prompts in supplementary.

to some extent in region-based methods (Avrahami et al., 2023; Nitzan et al., 2024; Kahatapitiya et al., 2024), but not sufficiently in the context of video generation.

Motivated by the above, we introduce Adaptive Caching (*AdaCache*) for accelerating video diffusion transformers. This approach requires *no training* and can seamlessly be integrated into a baseline video DiT at inference, as a plug-and-play component. The core idea of our proposal is to cache residual computations within transformer blocks (*e.g.* attention or MLP outputs) in a certain diffusion step, and reuse them through a number of subsequent steps, that is dependent on the video being generated. We do this by devising a caching schedule, *i.e.*, deciding when-to-recompute-next whenever making a residual computation. This decision is guided by a distance metric that measures the rate-of-change between previously-stored and current representations. If the distance is high we would not cache for an extended period (*i.e.*, #steps), to avoid reusing incompatible representations. We further introduce a Motion Regularization (*MoReg*) to allocate computations based on the motion content in the video being generated. This is inspired by the observation that high-moving sequences require more diffusion steps to achieve a reasonable quality. Altogether, our pipeline is applied on top of multiple video DiT baselines showing much-faster inference speeds without sacrificing generation quality (see Fig. 1). Finally, we validate the effectiveness of our contributions and justify our design decisions through ablations and qualitative comparisons.

## 2 RELATED WORK

**Diffusion-based Video Generation** (Singer et al., 2022; Ho et al., 2022; Blattmann et al., 2023a; Girdhar et al., 2023; Chen et al., 2024a) has surpassed the quality and diversity of GAN-based approaches (Vondrick et al., 2016; Saito et al., 2017; Tulyakov et al., 2018; Clark et al., 2019; Yu et al., 2022), while also being competitive with recent Auto-Regressive models (Yan et al., 2021; Hong et al., 2022; Villegas et al., 2022; Kondratyuk et al., 2023; Xie et al., 2024). They have become a standard component in the pipelines for frame interpolation (Wang et al., 2024c; Feng et al., 2024), video outpainting (Fan et al., 2023; Chen et al., 2024e; Wang et al., 2024a), image-to-video Guo et al. (2023); Blattmann et al. (2023a); Xing et al. (2023), video-to-video (*i.e.*, video editing or translation) (Yang et al., 2023a; Yatim et al., 2024; Hu et al., 2024), personalization (Wu et al., 2024; Men et al., 2024), motion customization (Zhao et al., 2023a; Xu et al., 2024) and

compositional generation (Liu et al., 2022; Yang & Wang, 2024). The underlying architecture of video diffusion models has evolved from classical UNets (Ronneberger et al., 2015; Rombach et al., 2022) with additional spatio-temporal attention layers (He et al., 2022; Blattmann et al., 2023b; Chen et al., 2023b; Girdhar et al., 2023), to fully-fledged transformer-based (*i.e.*, DiT (Peebles & Xie, 2023)) architectures (Lu et al., 2023; Ma et al., 2024b; Gao et al., 2024; Zhang et al., 2024b). In the process, the latency of denoising (Song et al., 2020; Lu et al., 2022) has also scaled with larger models (Podell et al., 2023; Gao et al., 2024). This becomes critical especially in applications such as long-video generation (Yin et al., 2023; Wang et al., 2023a; Zhao et al., 2024a; Henschel et al., 2024; Tan et al., 2024; Zhou et al., 2024), while also affecting the growth of commercially-served video models (Runway AI, 2024; OpenAI, 2024; Luma AI, 2024; Kling AI, 2024).

**Efficiency of Diffusion models** has been actively explored with respect to both training and inference pipelines. Multi-stage training at varying resolutions (Chen et al., 2023a; 2024b; Gao et al., 2024) and high-quality data curation (Ramesh et al., 2022; Ho et al., 2022; Dai et al., 2023; Blattmann et al., 2023a) have cut down training costs significantly. In terms of inference acceleration, there exist two main approaches: (1) methods that require re-training such as step-distillation (Salimans & Ho, 2022; Meng et al., 2023; Sauer et al., 2023; Liu et al., 2023b), consistency regularization (Song et al., 2023; Luo et al., 2023), quantization (Li et al., 2023; Chen et al., 2024c; He et al., 2024; Wang et al., 2024b; Deng et al., 2024), and architecture search/compression (Zhao et al., 2023b; Yang et al., 2023b; Li et al., 2024b), or (2) methods that require no re-training such as token reduction (Bolya & Hoffman, 2023; Li et al., 2024a; Kahatapitiya et al., 2024) and caching (Ma et al., 2024c; Wimbauer et al., 2024; Habibian et al., 2024; Chen et al., 2024d; Zhao et al., 2024b). Among these, training-free methods are more-attractive as they can be widely-adopted without any additional costs. This becomes especially relevant for video diffusion models that are both expensive to train and usually very slow at inference. In this paper, we explore a caching-based approach tailored for video DiTs. Different from prior fixed caching schedules in UNet-based (Ma et al., 2024c; Wimbauer et al., 2024; Habibian et al., 2024) and DiT-based (Chen et al., 2024d; Zhao et al., 2024b) pipelines, we introduce a content-dependent (*i.e.*, adaptive) caching scheme to squeeze out the best quality-latency trade-off.

**Content-adaptive Generation** may focus on improving consistency (Couairon et al., 2022; Bar-Tal et al., 2022; Avrahami et al., 2022; 2023; Wang et al., 2023b; Xie et al., 2023), quality (Suin et al., 2024; Abu-Hussein et al., 2022), and/or efficiency (Tang et al., 2023; Nitzan et al., 2024; Kahatapitiya et al., 2024; Starodubcev et al., 2024). Most region-based methods (*e.g.* image or video editing) rely on a user-provided mask to ensue consistent generations aligned with context information (Avrahami et al., 2023; Xie et al., 2023). Some others automatically detect (Suin et al., 2024) or retrieve (Abu-Hussein et al., 2022) useful information to improve generation quality. Among efficiency-oriented approaches, there exist proposals for selectively-processing a subset of latents (Nitzan et al., 2024; Kahatapitiya et al., 2024), switching between diffusion models with varying compute budgets (Starodubcev et al., 2024), or adaptively-controlling the number of denoising steps (Tang et al., 2023; Wimbauer et al., 2024). AdaDiff (Tang et al., 2023) skips all subsequent computations in a denoising step, if an uncertainty threshold is met at a certain layer. Block caching (Wimbauer et al., 2024) introduces a caching schedule tailored for a given pretrained diffusion model. Both these handle image generation tasks. In contrast, our proposed AdaCache— which also controls #denoising-steps adaptively— provides better flexibility, and is applied to more-challenging video generation. It is flexible in the sense that (1) it can selectively-cache any layer or even just a specific module within a layer, and (2) it is tailored to each video generation instead of being fixed for a given architecture. Thus, AdaCache gains more control over the diffusion process, enabling a better-adaptive compute allocation.

## 3 NOT ALL VIDEOS ARE CREATED EQUAL

In this section, we motivate the need for a content-dependent denoising process, and show how it can help maximize the quality-latency trade-off. This motivation is based on a couple of interesting observations which we describe below.

First, we note that each video is unique. Hence, videos have varying levels of complexity. Here, the complexity of a given video can be expressed by the rate-of-change of information across both space and time. Simpler videos may contain more homogeneous regions and/or static content. In contrast,

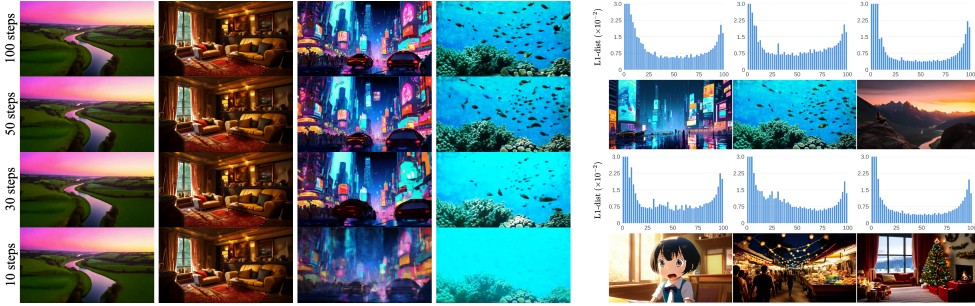

Figure 2: **Not all videos are created equal:** We show frames from 720p - 2s video generations based on Open-Sora (Zheng et al., 2024). (Left) We try to break each generation by reducing the number of diffusion steps. Interestingly, not all videos have the same break point. Some sequences are extremely robust (*e.g.* first-two columns), while others break easily. (Right) When we plot the difference between computed representations in subsequent diffusion steps, we see unique variations (L1-dist vs. #steps). If we are to reuse similar representations, it needs to be tailored to each video. Both these observations suggest the need for a content-dependent denoising process, which is the founding motivation of Adaptive Caching. Best-viewed with zoom-in. Prompts in supplementary.

complex videos have more high-frequency details and/or significant motion. Standard video compression techniques exploit such information to achieve best possible compression ratios without sacrificing quality (Wiegand et al., 2003; Sullivan et al., 2012). Motivated by the same, we explore how the compute cost affects the quality of video generations based on DiTs. We measure this w.r.t. the number of denoising steps, and the observations are shown in Fig. 2 (Left). Some video sequences are very robust, and achieve reasonable quality even at fewer denoising steps. Others break easily when we keep reducing the #steps, but the break point varies. This observation suggests that the minimal #steps (or, computations) required to generate a video with a reasonable quality varies, and having a content-dependent denoising schedule can exploit this to achieve the best speedups.

Next, we observe how the computed representations (*i.e.*, residual connections in attention or MLP blocks within DiT) change during the denoising process, across different video generations. This may reveal the level of compute redundancy in each video generation, enabling us to reuse representations and improve efficiency. More specifically, we visualize the feature differences between subsequent diffusion steps as histograms given in Fig. 2 (Right). Here, we report L1-distance vs. #steps. We observe that each histogram is unique. Despite having higher changes in early/latter steps and smaller changes in the middle, the overall distribution and the absolute values vary considerably. A smaller change corresponds to higher redundancy across subsequent computations, and an opportunity for reusing. This motivates the need for a non-uniform compute-schedule not only within the diffusion process of a given video (*i.e.*, at different stages of denoising), but also across different videos.

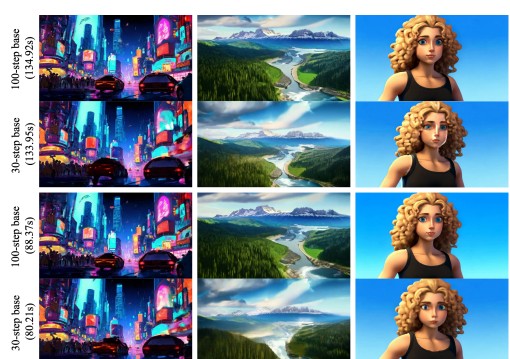

Figure 3: **Videos generated at a capped-budget:** There exist different configurations for generating videos at an approximately-fixed latency (*e.g.* having arbitrary #denoising-steps, yet only computing a fixed #representations and reusing otherwise). We observe a significant variance in quality in such videos. Best-viewed with zoom-in. Prompts in supplementary.

Finally, we evaluate the video generation quality at a capped-budget (*i.e.*, fixed computations or latency). We can have multiple generation configurations at an approximately-fixed latency, by computing a constant number of representations. For instance, we can cache and reuse representations more-frequently in a setup with more denoising steps, still having the same latency of a process with fewer steps. The observations of a study with either 30 or 100 base denoising steps is shown in Fig. 3. We see that the generation quality varies significantly despite spending a similar cost and having the same underlying pretrained DiT. This motivates us to think about how best to allocate our resources at inference, tailored for each video generation.

# 4 ADAPTIVE CACHING FOR FASTER VIDEO DiTs

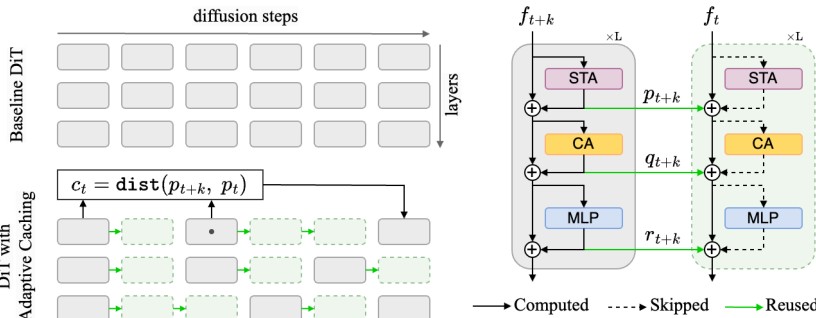

Figure 4: **Overview of Adaptive Caching:** (Left) During the diffusion process, we choose to cache residual computations within selected DiT blocks. The caching schedule is *content-dependent*, as we decide when to compute the next representation based on a distance metric ($c_t$). This metric measures the rate-of-change from previously-computed (and, stored) representation to the current one, and can be evaluated per-layer or the DiT as a whole. Each computed residual can be cached and reused across multiple steps. (Right) We only cache the residuals (*i.e.*, skip-connections) which amount to the actual computations (*e.g.* spatial-temporal/cross attention, MLP). The iteratively denoised representation (*i.e.*, $f_{t+k}$, $f_t$) always gets updated either with computed or cached residuals.

## 4.1 PRELIMINARIES: VIDEO DIFFUSION TRANSFORMERS

Video Diffusion Transformers are extended from Latent Diffusion Transformers (DiTs) (Peebles & Xie, 2023) introduced for image generation. DiTs provide a much-more streamlined, scalable architecture compared to prior UNet-based diffusion models (Rombach et al., 2022), by only having transformer blocks with a homogeneous token resolution (instead of convolutional blocks with up/downsampling). A simplified transformer block (*i.e.*, w/o normalizing or timestep conditioning layers) in a video DiT is shown in Fig. 4 (right)— gray block. It consists of spatial-temporal attention (STA), cross-attention (CA) and linear (MLP) layers. Depending on the implementation, STA may be a single joint spatio-temporal attention layer, or separate spatial and temporal attention layers repeated within alternating blocks. Without loss of generality, let us denote a latent feature at the input/output of such block by $f_t^l$ and $f_t^{l+1}$, respectively. Here, $l$ represents the layer index, and $t$, the diffusion timestep. A simplified flow of computations within each block can be represented as,

$$p_t^l = \texttt{STA}(f_t^l) \; ; \quad \tilde{f}_t^l = f_t^l + p_t^l \; , \tag{1}$$

$$q_t^l = \texttt{CA}(\tilde{f}_t^l) \quad ; \quad \bar{f}_t^l = \tilde{f}_t^l + q_t^l \; , \tag{2}$$

$$r_t^l = \texttt{MLP}(\bar{f}_t^l) \; ; \quad f_t^{l+1} = \bar{f}_t^l + r_t^l \; . \tag{3}$$

Here $p_t^l$, $q_t^l$ and $r_t^l$ are residual connections corresponding to each compute-element. Such computations repeat through $L$ layers, generating the noise prediction of each step $t$, and across a total of $T$ denoising steps. In the current streamlined video DiT architectures with homogeneous token resolutions, each layer of each denoising step costs the same.

## 4.2 ADAPTIVE CACHING

In this subsection, we introduce Adaptive Caching (*AdaCache*), a *training-free* mechanism for content-dependent compute allocation in video DiTs. The overview of Adaptive Caching is shown in Fig. 4. Compared to a standard DiT that computes representations for all layers across all diffusion steps, in AdaCache, we decide which layers or steps to compute, adaptively (*i.e.*, dependent on each video generation). This decision is based on the rate-of-change in the residual connections (*e.g.* $p_t^l$, $q_t^l$ or $r_t^l$) across diffusion steps, which amount to all significant computations within the DiT. Without loss of generality, let us assume that the residuals in block $l$ in current and immediately-prior diffusion steps $t$ and $t+k$ are already computed. Here, step $t+k$ is identified as 'immediately-prior'

to step $t$ since any residuals between these two steps are not computed (*i.e.*, cached residuals reused). We make a decision on the next compute-step based on the distance metric ($c_t^l$) given by,

$$c_t^l = \texttt{dist}(p_{t+k}^l, \ p_t^l) = \|p_t^l - p_{t+k}^l\| \ / \ k \ . \tag{4}$$

Here, we use L1 distance by default, but other distance metrics can also be applied (*e.g.* L2, cosine). Once we have the distance metric, we select the next caching rate ($\tau_t^l$) based on a pre-defined codebook of basis cache-rates that corresponds to the original denoising schedule (*i.e.*, #steps). The codebook is basically a collection of cache-rates coupled with metric thresholds to select them.

$$\tau_t^l = \texttt{codebook}(c_t^l) \ . \tag{5}$$

For all denoising steps within $t$ and $t - \tau$, we reuse previously-cached representations and only recompute after the current caching schedule (while also estimating the metric, again).

$$p_{t-k}^l = \begin{cases} p_t^l & \text{if} \quad k < \tau_t^l; \\ p_{t-k}^l = \texttt{STA}(f_{t-k}^l) & \text{if} \quad k = \tau_t^l. \end{cases} \tag{6}$$

The same applies to other residual computations (*e.g.* $q_{t-k}^l$, $r_{t-k}^l$) as well. By design, we can have unique caching schedules for each layer (and, each residual computation). However, we observe that it will make the generations unstable. Therefore, we decide to have a common metric (*i.e.*, $c_t^l = c_t$) and hence, a common caching rate (*i.e.*, $\tau_t^l = \tau_t$) across all DiT layers. For instance, we can consider an averaged metric, or a metric computed at a certain layer to decide the caching schedule. Meaning, when we recompute residuals in a certain step, we do so for the whole DiT rather than selectively for each layer.

Overall, this setup allows us to adaptively-control the compute spent on each video generation. If the rate-of-change between residuals is high, we will have a smaller caching rate, and otherwise, we have a higher rate. The choice of a lightweight distance metric (*e.g.* L1) helps us avoid any additional latency overheads.

### 4.3 MOTION REGULARIZATION

To further improve Adaptive Caching by making use of video information, we introduce a Motion Regularization (*MoReg*). This is motivated by the observation that the optimal number of denoising steps varies based on the motion content of each generated video. The core idea is to cache less (*i.e.*, recompute more) if a generated video has a high motion content. To regularize our caching schedule, we estimate a latent motion-score ($m_t^l$) based on residual frame differences. Without loss of generality, let us denote residual latent frames of $p_t^l$ as $\{p_{t, n}^l \mid n = 0, \cdots, N - 1\}$ where $N$ is the #frames in latent space (given by the VAE encoder). We estimate the motion-score as,

$$m_t^l = \|p_{t, \ i:N}^l - p_{t, \ 0:N-i}^l\| \ . \tag{7}$$

Here, $i$ denotes the frame step-size (or, frame-rate), $\| \cdot \|$, the L1 distance, and $i : j$, all frames within the corresponding range. However, since we operate on noisy-latents, we observe that our motion estimate in early diffusion steps is not reliable. Meaning, it does not provide a reasonable regularization in early steps (*i.e.*, the change in caching schedule does not correlate well with the observed motion of a generated video in pixel space). To alleviate this, we also compute a motion-gradient ($mg_t^l$) across diffusion steps, which can act as a reasonable early-predictor of motion that we may observe in latter diffusion steps (that also correlates with the motion in pixel space).

$$mg_t^l = (m_t^l - m_{t+k}^l) \ / \ k \ . \tag{8}$$

Finally, we use both motion and motion-gradient as a multiplier for the distance metric ($c_t^l$) to regularize our caching schedule.

$$c_t^l = c_t^l \cdot (m_t^l + mg_t^l) \ . \tag{9}$$

This means, when we have a higher estimated motion, the distance metric will be increased and a smaller basis cache-rate will be selected from the codebook. Similar to before, we enforce a common motion-regularization to all DiT layers by computing a common motion score (*i.e.*, $m_t^l = m_t$, $mg_t^l = mg_t$), ensuring the stability of denoising process. We can also choose to compute motion at different frame-rates, which we ablate in our experiments.

Table 1: **Quantitative evaluation of quality and latency:** Here, we compare AdaCache with other *training-free* DiT acceleration methods (*e.g.* Δ-DiT (Chen et al., 2024d), T-GATE (Zhang et al., 2024a), PAB (Zhao et al., 2024b)) on mutliple video baselines (*e.g.* Open-Sora (Zheng et al., 2024) 480p - 2s at 30-steps, Open-Sora-Plan (Lab & etc., 2024) 512×512 - 2.7s at 150-steps, Latte (Ma et al., 2024b) 512×512 - 2s at 50-steps). We measure the generation quality with VBench Huang et al. (2024), PSNR, LPIPS and SSIM, while reporting complexity with FLOPs, latency and speedup (measured on a single 80G A100 GPU). AdaCache-fast consistently shows the best speedups at a comparable or slightly-lower generation quality. AdaCache-slow gives absolute-best quality while still being faster than prior methods. Our motion-regularization significantly improves the generation quality consistently, with a minimal added-latency.

| Method | VBench (%) ↑ | PSNR ↑ | LPIPS ↓ | SSIM ↑ | FLOPs (T) | Latency (s) | Speedup |
|---|---|---|---|---|---|---|---|
| Open-Sora Zheng et al. (2024) | 79.22 | – | – | – | 3230.24 | 54.02 | 1.00× |
| + Δ-DiT (Chen et al., 2024d) | 78.21 | 11.91 | 0.5692 | 0.4811 | 3166.47 | – | – |
| + T-GATE (Zhang et al., 2024a) | 77.61 | 15.50 | 0.3495 | 0.6760 | 2818.40 | – | – |
| + PAB-fast (Zhao et al., 2024b) | 76.95 | 23.58 | 0.1743 | 0.8220 | 2558.25 | 40.23 | 1.34× |
| + PAB-slow (Zhao et al., 2024b) | 78.51 | 27.04 | 0.0925 | 0.8847 | 2657.70 | 44.93 | 1.20× |
| + AdaCache-fast | 79.39 | 24.92 | 0.0981 | 0.8375 | **1331.97** | **24.16** | **2.24×** |
| + AdaCache-fast (w/ MoReg) | 79.48 | 25.78 | 0.0867 | 0.8530 | 1383.66 | 25.71 | 2.10× |
| + AdaCache-slow | **79.66** | **29.97** | **0.0456** | **0.9085** | 2195.50 | 37.01 | 1.46× |
| Open-Sora-Plan (Lab & etc., 2024) | 80.39 | – | – | – | 12032.40 | 129.67 | 1.00× |
| + Δ-DiT (Chen et al., 2024d) | 77.55 | 13.85 | 0.5388 | 0.3736 | 12027.72 | – | – |
| + T-GATE (Zhang et al., 2024a) | 80.15 | 18.32 | 0.3066 | 0.6219 | 10663.32 | – | – |
| + PAB-fast (Zhao et al., 2024b) | 71.81 | 15.47 | 0.5499 | 0.4717 | 8551.26 | 89.56 | 1.45× |
| + PAB-slow (Zhao et al., 2024b) | 80.30 | 18.80 | 0.3059 | 0.6550 | 9276.57 | 98.50 | 1.32× |
| + AdaCache-fast | 75.83 | 13.53 | 0.5465 | 0.4309 | **3283.60** | **35.04** | **3.70×** |
| + AdaCache-fast (w/ MoReg) | 79.30 | 17.69 | 0.3745 | 0.6147 | 3473.68 | 36.77 | 3.53× |
| + AdaCache-slow | **80.50** | **22.98** | **0.1737** | **79.10** | 4983.30 | 58.88 | 2.20× |
| Latte (Ma et al., 2024b) | 77.40 | – | – | – | 3439.47 | 32.45 | 1.00× |
| + Δ-DiT (Chen et al., 2024d) | 52.00 | 8.65 | 0.8513 | 0.1078 | 3437.33 | – | – |
| + T-GATE (Zhang et al., 2024a) | 75.42 | 19.55 | 0.2612 | 0.6927 | 3059.02 | – | – |
| + PAB-fast (Zhao et al., 2024b) | 73.13 | 17.16 | 0.3903 | 0.6421 | 2576.77 | 24.33 | 1.33× |
| + PAB-slow (Zhao et al., 2024b) | 76.32 | 19.71 | 0.2699 | 0.7014 | 2767.22 | 26.20 | 1.24× |
| + AdaCache-fast | 76.26 | 17.70 | 0.3522 | 0.6659 | **1010.33** | **11.85** | **2.74×** |
| + AdaCache-fast (w/ MoReg) | 76.47 | 18.16 | 0.3222 | 0.6832 | 1187.31 | 13.20 | 2.46× |
| + AdaCache-slow | **77.07** | **22.78** | **0.1737** | **0.8030** | 2023.65 | 20.35 | 1.59× |

## 5 EXPERIMENTS

### 5.1 IMPLEMENTATION DETAILS

We select multiple prominent open-source video DiTs as backbone video generation pipelines in our experiments, namely, Open-Sora-v1.2 (Zheng et al., 2024), Open-Sora-Plan-v1.1 (Lab & etc., 2024) and Latte (Ma et al., 2024b). Since we focus on inference-based latency optimizations (*i.e.*, without any re-training), we compare AdaCache against similar methods such as Δ-DiT (Chen et al., 2024d), T-GATE (Zhang et al., 2024a) and PAB (Zhao et al., 2024b). In our main experiments, we generate 900+ videos based on standard VBench (Huang et al., 2024) benchmark prompts at the corresponding generation settings of each baseline (*e.g.* 480p - 2s with 30-steps in Open-Sora, 512×512 - 2.7s with 150-steps in Open-Sora-Plan and 512×512 - 2s with 50-steps in Latte) measuring multiple quality-complexity metrics. We report VBench average and reference-based PSNR, SSIM and LPIPS as quality metrics, and report FLOPs, Latency (s) and Speedup as complexity metrics. Here, Latency is measured on a single 80G A100 GPU. In all our ablations and qualitative results, we experiment on the standard prompts from Open-Sora benchmark gallery, generating 720p - 2s videos with 100-steps.

### 5.2 MAIN RESULTS

In Table 1, we present a quantitative evaluation of quality and latency on VBench (Huang et al., 2024) benchmark. We consider three variants of AdaCache: a slow variant, a fast variant with more speedup and the same with motion regularization. We compare with other training-free acceleration methods, showing consistently better speedups with a comparable generation quality. With Open-Sora (Zheng et al., 2024) baseline, AdaCache-slow outperforms others on all quality metrics, while giving a 1.46× speedup compared to PAB (Zhao et al., 2024b) with 1.20× speedup. AdaCache-fast

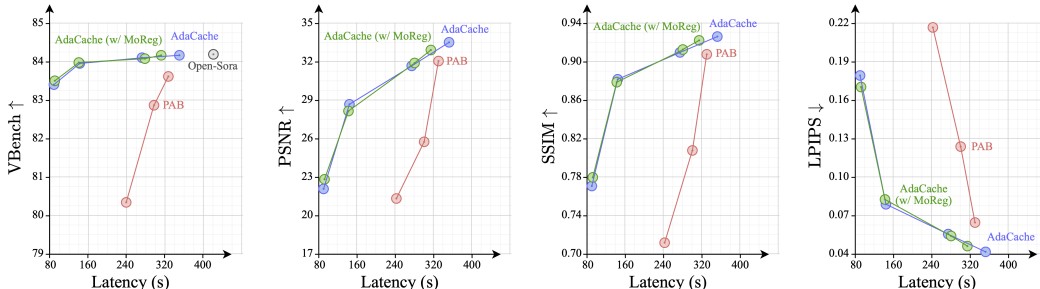

Figure 5: **Quality-Latency trade-off:** We show quality vs. latency curves for different configurations of AdaCache and PAB (Zhao et al., 2024b), with Open-Sora (Zheng et al., 2024) 720p - 2s generations. AdaCache outperforms PAB consistently, showing a more-stable performance while reducing latency. This stability is more-prominent in reference-free metric VBench (Huang et al., 2024) compared to reference-based metrics, validating that AdaCache generations are aligned with human preference even at its fastest speeds, despite not being exactly-aligned with the reference.

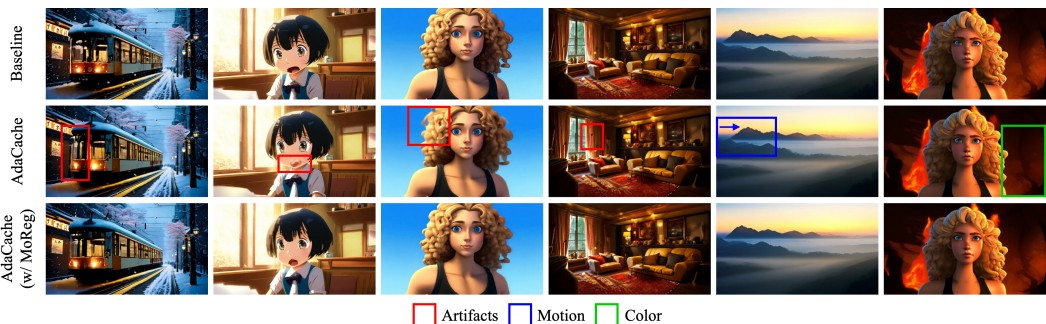

Figure 6: **Visualizing the impact of Moiton Regularization:** We show a qualitative comparison of AdaCache and AdaCache (w/ MoReg), applied on top of Open-Sora (Zheng et al., 2024) baseline. Here, we consider generation of 720p - 2s clips. Despite giving a 4.7× speedup, AdaCache can also introduce some inconsistencies over time (*e.g.* artifacts, motion, color). Motion Regularization helps avoid most of them by allocating more computations proportional to the amount of motion (still giving a 4.5× speedup). Best-viewed with zoom-in. Prompts in supplementary.

gives the highest acceleration of 2.24× with a slight drop in quality. AdaCache-fast (w/ MoReg) shows a clear improvement in quality compared to AdaCache-fast, validating the effectiveness of our regularization and giving a comparable speedup of 2.10×. All AdaCache variants outperform even the baseline (w/o any acceleration) on VBench average quality, which aligns better with human preference compared to other reference-based metrics. Similar observations hold with the other baselines as well. With Open-Sora-Plan (Lab & etc., 2024), AdaCache shows the best speedup of 3.70× compared to the previous-best 1.45× of PAB, and the best quality with a 2.20× speedup. With Latte (Ma et al., 2024b), we gain the best speedup of 2.74× compared to prior 1.33×, and the best overall quality with a 1.59× speedup.

## 5.3 ABLATION STUDY

**Quality-Latency trade-off:** In Fig. 5, we compare the quality-latency trade-off of AdaCache with PAB (Zhao et al., 2024b). First, we note that AdaCache enables significantly higher reduction rates (*i.e.*, much-smaller absolute latency) compared to PAB. Moreover, across this whole range of latency configurations, AdaCache gives a more-stable performance over PAB, on all quality metrics. Such behavior is especially evident in reference-free metric VBench (Huang et al., 2024), that aligns better with human preference. Even if we see a drop in reference-based scores (*e.g.* PSNR, SSIM) at extreme reduction rates, the qualitative results suggest that the generations are still good (see Fig. 1), despite not being aligned exactly with the reference.

Table 2: **Ablation study:** We evaluate different design decisions of AdaCache on Open-Sora (Zheng et al., 2024) benchmark prompts, reporting VBench (Huang et al., 2024) scores (%), latency (s) and speedup. Here, we consider 32 videos generated with 100 diffusion steps, and use VBench custom dataset evaluation as suggested in the benchmark.

(a) **AdaCache with Motion Regularization**: We show different variants of AdaCache. All versions achieve significant speedups. AdaCache + MoReg shows a better quality with a slightly-lower speedup.

| Method | VBench | Latency | Speedup |
|---|---|---|---|
| Open-Sora (Zheng et al., 2024) | 84.16 | 419.60 | 1.0× |
| + AdaCache | 83.40 | 89.53 | 4.7× |
| + AdaCache + MoReg | 83.50 | 93.50 | 4.5× |
| + AdaCache + MoReg (w/o grad) | 83.36 | 89.01 | 4.7× |
| + AdaCache + MoReg (multi-step) | 83.42 | 95.65 | 4.4× |

(b) **Speedups at different resolutions**: We compare AdaCache with baselines at different resolutions. AdaCache generalizes across resolutions, providing a stable acceleration.

| Resolution | AdaCache | VBench | Latency | Speedup |
|---|---|---|---|---|
| 480p - 2s | ✗ | 83.68 | 173.84 | 1.0× |
| | ✓ | 83.18 | 38.52 | 4.5× |
| 480p - 4s | ✗ | 82.77 | 349.90 | 1.0× |
| | ✓ | 82.16 | 80.16 | 4.4× |
| 720p - 2s | ✗ | 84.16 | 419.60 | 1.0× |
| | ✓ | 83.40 | 89.53 | 4.7× |

(c) **Cache metric**: Among different caching metrics, L1/L2 give similar performance compared to cosine distance.

| Cache metric | VBench | Latency |
|---|---|---|
| L1-distance | 83.40 | 89.53 |
| L2-distance | 83.50 | 92.70 |
| Cosine-distance | 83.19 | 86.74 |

(d) **Cache location**: We compute the cache metric at mid-DiT, for best quality-latency trade-off.

| Cache loc. | VBench | Latency |
|---|---|---|
| Start | 83.30 | 87.55 |
| Mid | 83.40 | 89.53 |
| End | 83.43 | 91.20 |
| Multiple | 83.41 | 90.27 |

(e) **AdaCache Variants**: We achieve a range of speedups (and quality) by controlling the basis cache-rates in AdaCache. Our default configuration is AdaCache-fast.

| AdaCache variant | Basis-rates | VBench | Latency |
|---|---|---|---|
| AdaCache-fast | 12-10-8-6-4-3 | 83.40 | 89.53 |
| AdaCache-mid | 8-6-4-2-1 | 83.94 | 143.87 |
| AdaCache-slow | 2-1 | 84.12 | 274.30 |

**AdaCache with Motion Regularization:** We compare AdaCache with different versions of motion regularization in Table 2a. Both vanilla and motion-regularized versions provide significant speedups, 4.7× and 4.5× respectively, at a comparable quality with baseline Open-Sora (Zheng et al., 2024). Considering motion-gradient as an early-prediction of motion at latter diffusion steps helps (83.50 vs. 83.36 on VBench). We also estimate motion at different frame-rates by considering varying step-size in frame differences, which seems to increase the latency without improving quality. Overall, we consider AdaCache + MoReg as the confifuration with best quality-latency trade-off. This improvement in quality is more-prominent in qualitative examples shown in Fig. 6 and benchmark comparison in Table 1.

**Speedups at different resolutions:** In Table 2b, we compare the trade-offs of AdaCache at various resolutions of video generations, namely, 480p - 2s, 480p - 4s and 720p - 2s, all at 100-steps. AdaCache provides consistent speedups across different resolutions without affecting the quality.

**Cache metric and location:** When adaptively deciding the caching schedule, we consider different metrics to compute the rate-of-change between representations, namely, L1/L2 distance or cosine distance. Among these, L1/L2 give an absolute measure which aligns better with the actual change. In contrast, cosine computes a normalized-distance, which is not a good estimate of change (*e.g.* if the representations differ only by a scale, the distance will be zero, even though we want to have a non-zero metric). This observation is verified by the results in Table 2c. Moreover, we consider computing the cache metric at various locations (*i.e.*, layers) in the DiT. Doing so at a single layer (*e.g.* start, mid, end) is not significantly different from computing an aggregate over multiple-layers (see Table 2d). By default, we compute the cache metric in the mid-layer as a reasonable choice without extra overheads.

**AdaCache variants:** To achieve a range of speedups (and quality), we consider different basis cache-rates in our AdaCache implementation. For instance, we can have higher-speedup with a slightly-lower quality (*e.g.* AdaCache-fast), a lower-speedup with a higher-quality (*e.g.* AdaCache-slow), or balance both (*e.g.* AdaCache-mid). We can conveniently control this by having corresponding basis cache-rates as shown in Table 2e. By defualt, we resort to AdaCache-fast which gives the best speedups.

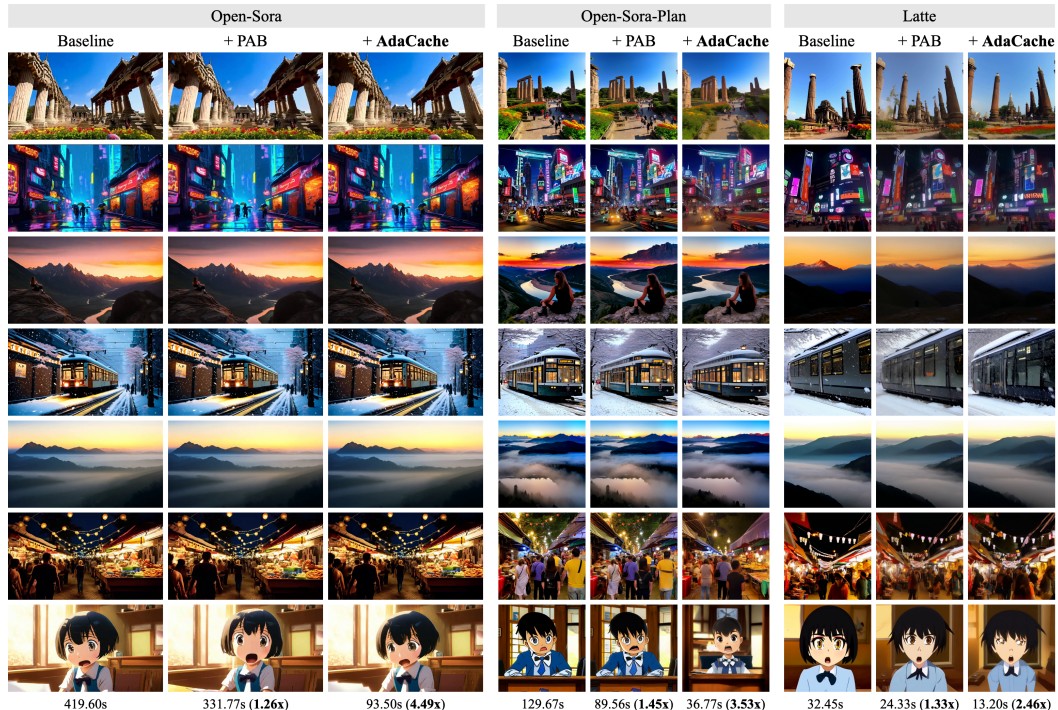

Figure 7: **Qualitative comparison:** We show qualitative results on multiple video-DiT baselines including Open-Sora (Zheng et al., 2024) (720p - 2s at 100-steps), Open-Sora-Plan (Lab & etc., 2024) (512×512 - 2.7s at 150-steps) and Latte (Ma et al., 2024b) (512×512 - 2s at 50-steps), while comparing against prior *training-free* inference acceleration method PAB (Zhao et al., 2024b). Ada-Cache shows a comparable generation quality at much-faster speeds. Best-viewed with zoom-in. Prompts in supplementary.

## 5.4 QUALITATIVE RESULTS

In Fig. 7, we present qualitative results on mutliple video DiT baselines, including Open-Sora (Zheng et al., 2024), Open-Sora-Plan (Lab & etc., 2024) and Latte (Ma et al., 2024b). We compare AdaCache against each baseline and prior training-free inference acceleration method for DiTs, PAB (Zhao et al., 2024b). Here, we consider three different configurations: 720p - 2s generations at 100-steps for Open-Sora, 512×512 - 2.7s generations at 150-steps for Open-Sora-Plan, and 512×512 - 2s generations at 50-steps for Latte, while considering standard prompts from Open-Sora gallery (see supplementary for prompt details). AdaCache shows comparable generation quality, while having much-faster inference pipelines. In fact, it achieves 4.49× (vs. 1.26× in PAB), 3.53× (vs. 1.45× in PAB), 2.46× (vs. 1.33× in PAB) speedups respectively on the three considered baseline DiTs. In most cases our generations are aligned well with the baseline in the pixel-space. Yet this is not a strict requirement, as the denoising process can deviate considerably from that of the baseline, at high caching rates. Still, AdaCache is faithful to the text prompt and is not affected by significant artifacts.

## 6 CONCLUSION

In this paper, we introduced Adaptive Caching (*AdaCache*), a plug-and-play component that improves the the inference speed of video generation pipelines based on diffusion transformers, without needing any re-training. It caches residual computations, while also devising the caching schedule dependent on each video generation. We further proposed a Motion Regularization (*MoReg*) to utilize video information and allocate computations based on motion content, improving the quality-latency trade-off. We apply our contributions on multiple open-source video DiTs, showing comparable generation quality at a fraction of latency. We believe AdaCache is widely-applicable with minimal effort, helping democratize high-fidelity long video generation.

REPRODUCIBILITY STATEMENT

We use open-source video DiTs (w/ publicly-available code and pretrained-weights) in all our experiments. As we rely on zero-shot (*i.e.*, *training-free*) inference acceleration, we do not update pretrained weights. All our quantitative evaluations and generated videos correspond to standard benchmark prompts that are also publicly-available. Our method details all required steps to reproduce the proposed contributions. Finally, we pledge to release our code together with the paper to support further research.

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

# A APPENDIX

## A.1 TEXT PROMPTS USED IN QUALITATIVE EXAMPLES

Text prompts corresponding to the video generations in Fig. 1:

- A Japanese tram glides through the snowy streets of a city, its
  sleek design cutting through the falling snowflakes with grace.
  The tram's illuminated windows cast a warm glow onto the snowy
  surroundings, creating a cozy atmosphere inside. Snowflakes dance
  in the air, swirling around the tram as it moves along its tracks.
  Outside, the city is blanketed in a layer of snow, transforming
  familiar streets into a winter wonderland. Cherry blossom trees,
  now bare, stand quietly along the tram tracks, their branches
  dusted with snow. People hurry along the sidewalks, bundled up
  against the cold, while the tram's bell rings softly, announcing
  its arrival at each stop.

- a picturesque scene of a tranquil beach at dawn. the sky is
  painted in soft pastel hues of pink and orange, reflecting on the
  calm, crystal-clear water. gentle waves lap against the sandy
  shore, where a lone seashell lies near the water's edge. the
  horizon is dotted with distant, low-lying clouds, adding depth to
  the serene atmosphere. the overall mood of the video is peaceful
  and meditative, with no text or additional objects present. the
  focus is on the natural beauty and calmness of the beach, captured
  in a steady, wide shot.

- a bustling night market scene with vibrant stalls on either side
  selling food and various goods. the camera follows a person
  walking through the crowded, narrow alley. string lights hang
  overhead, casting a warm, festive glow. people of all ages
  are talking, browsing, and eating, creating an atmosphere full
  of lively energy. occasional close-ups capture the details of
  freshly cooked dishes and colorful merchandise. the video is
  dynamic with a mixture of wide shots and close-ups, capturing the
  essence of the night market without any text or sound.

- a dynamic aerial shot showcasing various landscapes. the
  sequence begins with a sweeping view over a dense, green forest,
  transitioning smoothly to reveal a winding river cutting through
  a valley. next, the camera rises to capture a panoramic view of
  a mountain range, the peaks dusted with snow. the shot shifts to
  a coastal scene, where waves crash against rugged cliffs under a
  partly cloudy sky. finally, the aerial view ends over a bustling
  cityscape, with skyscrapers and streets filled with motion and
  life. the video does not contain any text or additional overlays.

- a cozy living room scene with a christmas tree in the corner
  adorned with colorful ornaments and twinkling lights. a fireplace
  with a gentle flame is situated across from a plush red sofa,
  which has a few wrapped presents placed beside it. a window
  to the left reveals a snowy landscape outside, enhancing the
  festive atmosphere. the camera slowly pans from the window to the
  fireplace, capturing the warmth and tranquility of the room. the
  soft glow from the tree lights and the fire illuminates the room,
  casting a comforting ambiance. there are no people or text in the
  video, focusing purely on the holiday decor and cozy setting.

Text prompts corresponding to new video generations in Fig. 2:

- a breathtaking aerial view of a river meandering through a lush green landscape. the river, appearing as a dark ribbon, cuts through the verdant fields and hills, reflecting the soft light of the pinkish-orange sky. the sky, painted in hues of pink and orange, suggests the time of day to be either sunrise or sunset. the landscape is dotted with trees and bushes, adding to the natural beauty of the scene. the perspective of the video is from above, providing a bird's eye view of the river and the surrounding landscape. the colors , the river, the landscape, and the sky all come together to create a serene and picturesque scene.

- A cozy living room, surrounded by soft cushions and warm lighting. Describe the scene in vivid detail, capturing the feeling of comfort and relaxation.

- a nighttime scene in a bustling city filled with neon lights and futuristic architecture. the streets are crowded with people, some dressed in high-tech attire and others in casual cyberpunk fashion. holographic advertisements and signs illuminate the area in vibrant colors, casting a glow on the buildings and streets. futuristic vehicles and motorcycles are speeding by, adding to the city's dynamic atmosphere. in the background, towering skyscrapers with intricate designs stretch into the night sky. the scene is filled with energy, capturing the essence of a cyberpunk world.

- a close-up shot of a vibrant coral reef underwater. various colorful fish swim leisurely around the corals, creating a lively scene. the lighting is natural and slightly subdued, emphasizing the deep-sea environment. soft waves ripple across the view, occasionally bringing small bubbles into the frame. the background fades into a darker blue, suggesting deeper waters beyond. there are no texts or human-made objects visible in the video.

- a neon-lit cityscape at night, featuring towering skyscrapers and crowded streets. the streets are bustling with people wearing futuristic attire, and vehicles hover above in organized traffic lanes. holographic advertisements are projected onto buildings, illuminating the scene with vivid colors. a light rain adds a reflective sheen to the ground, enhancing the cyberpunk atmosphere. the camera pans slowly through the scene, capturing the energy and technological advancements of the city. the video does not contain any text or additional objects.

- a breathtaking view of a mountainous landscape at sunset. the sky is painted with hues of orange and pink, casting a warm glow over the scene. the mountains, bathed in the soft light, rise majestically in the background, their peaks reaching towards the sky. in the foreground, a woman is seated on a rocky outcrop, her body relaxed as she takes in the vie w. she is dressed in a black dress and boots, her attire contrasting with the natural surroundings. her position on the rock provides a vantage point over a river that meanders through the valley below. the river, a ribbon of blue, winds its way through the landscape, adding a dynamic element to the scene. the woman's gaze is directed towards the river, suggesting a sense of contemplation or admiration for the beauty of nature. the video is taken from a high angle, looking down on the woman and the landscape. this perspective enhances the sense of depth and scale in the image, emphasizing the vastness of the mountains and the river.

- an animated scene featuring a young girl with short black hair and a bow tie, seated at a wooden desk in a warmly lit room.  natural light filters through a window, illuminating the girl's wide eyes and open mouth, conveying a sense of surprise or shock.  she is dressed in a blue shirt with a white collar and dark vest.  the room's inviting atmosphere is complemented by wooden furniture and a framed picture on the wall.  the animation style is reminiscent of japanese anime, characterized by vibrant colors and expressive character designs.

Text prompts corresponding to new video generations in Fig. 3:

- a realistic 3d rendering of a female character with curly blonde hair and blue eyes.  she is wearing a black tank top and has a neutral expression while facing the camera directly.  the background is a plain blue sky, and the scene is devoid of any other objects or text.  the character is detailed, with realistic textures and lighting, suitable for a video game or high-quality animation.  there is no movement or additional action in the video.  the focus is entirely on the character's appearance and realistic rendering.

Text prompts corresponding to new video generations in Fig. 6:

- a breathtaking aerial view of a misty mountain landscape at sunrise.  the sun is just beginning to peek over the horizon, casting a warm glow on the scene.  the mountains, blanketed in a layer of fog, rise majestically in the background.  the mist is so dense that it obscures the peaks of the mountains, adding a sense of mystery to the scene.  in the foregro und, a river winds its way through the landscape, its path marked by the dense fog.  the river appears calm, its surface undisturbed by the early morning chill.  the colors in the video are predominantly cool, with the blue of the sky and the green of the trees contrasting with the warm orange of the sunrise.  the video is taken from a high vantage point, p roviding a bird's eye view of the landscape. this perspective allows for a comprehensive view of the mountains and the river, as well as the fog that envelops them.  the video doe s not contain any text or human activity, focusing solely on the natural beauty of the landscape.  the relative positions of the objects suggest a vast, untouched wilderness.

- a 3d rendering of a female character with curly blonde hair and striking blue eyes.  she is wearing a black tank top and is standing in front of a fiery backdrop.  the character is looking off to the side with a serious expression on her face. the background features a fiery orange and red color scheme, suggesting a volcanic or fiery environment.  the lighting in the scene is dramatic, with the character's face illuminated by a soft light that contrasts with the intense colors of the background. there are no texts or other objects in the image.  the style of the image is realistic with a high level of detail, indicative of a high-quality 3d rendering.

Text prompts corresponding to new video generations in Fig. 7:

- a scenic shot of a historical landmark.  the landmark is an ancient temple with tall stone columns and intricate carvings. the surrounding area is lush with greenery and vibrant flowers. the sky above is clear and blue, with the sun casting a warm glow over the scene.  tourists can be seen walking around, taking

pictures and admiring the architecture. there is no text or additional objects in the video.

- a vibrant cyberpunk street scene at night. neon signs and holographic advertisements illuminate the narrow street, casting colorful reflections on the rain-slicked pavement. various characters, dressed in futuristic attire, move along the sidewalks while robotic street vendors sell their wares. towering skyscrapers with glowing windows dominate the background, creating a sense of depth. the camera takes a wide-angle perspective, capturing the bustling and lively atmosphere of the cyberpunk cityscape. there are no texts or other objects outside of the described scene.

