# OpenReview forum: "Adaptive Caching for Faster Video Generation with Diffusion Transformers"
_ICLR.cc/2025/Conference — Submitted to ICLR 2025_

### Official Review · Reviewer_g4jp · 2024-10-26

**Soundness:** 2
**Presentation:** 2
**Contribution:** 3
**Rating:** 3
**Confidence:** 4

**Summary:**

This paper presents AdaCache which accelerates video generation by caching residual computations and devising adaptive caching schedule without requiring re-training. It also introduces MoReg to optimize computation based on motion. This paper demonstrates the effectiveness of AdaCache across various open-source models.

**Strengths:**

1.A novel adaptive algorithm is designed to control the number of steps for reusing cached features.

**Weaknesses:**

1. typo need to be corrected:
    1. Line346: SSIM of AdaCache-slow should be 0.7910?
2. The methodology section lacks clarity in certain areas. For example, in Line 276, a pre-defined codebook is mentioned, and the reviewer wants to know how this codebook was pre-defined. Was it manually set? If so, what criteria or method were used to set it? Is there any further detailed explanation regarding it?
3. The qualitative comparison provided by the authors is insufficient and does not adequately demonstrate the superiority of AdaCache. Although AdaCache achieves a higher acceleration speedup, as shown in Figure 7, its results on Open-Sora-Plan (e.g., Rows 1, 2, 6, 7) and Latte (e.g., Rows 2, 4, 6, 7) are significantly worse than those of PAB and the Baseline. Based on the visual quality presented in Figure 7, the reviewer expresses concerns about the stability of AdaCache in terms of visual quality.

---


After reviewing the authors' latest response to Reviewer 5tUr's comments, the reviewer has decided to further lower the score. The primary concerns stem from:

The reliability of quantitative comparison data

(i) In the latest responses (3) and (4), the authors stated: "We follow the original inference settings suggested by the original contributors—in OpenSora, this is image- and text-conditioned generation." This implies that in Table 1, the experimental setup for OpenSora involves image- and text-conditioned generation. However, as far as the reviewer knows, PAB does not use reference images as image conditions by default in OpenSora. Yet, the performance metrics listed for PAB in Table 1—including fidelity metrics such as PSNR and SSIM, as well as reference-free metrics like VBench—are entirely identical to those reported in the PAB paper. The reviewer is uncertain whether these data were directly from the PAB paper. If so, the comparison would be unfair since PAB defaults to operating without image conditioning.

(ii) The sources of the Delta-Dit and TGATE data in Table 1 are unclear. Additionally, the FLOPS and visual quality results are unusual, showing almost no speedup.

Insufficient clarity in the experimental details presented in the paper, which may lead to misunderstandings

(i) The authors conducted experiments on three models: Open-Sora, Open-Sora-Plan, and Latte. Open-Sora-Plan and Latte are text-conditioned, while Open-Sora uses a text- and image-conditioned setup. The reviewer believes that such a unique configuration should be explicitly clarified in the paper.

(ii) In the rebuttal, the authors mentioned that for OpenSora, reducing the timesteps from 100 to 30 leads to a dramatic change in the speedup factor, from 4.5x to 2.24x. This significant variation should be emphasized and analyzed in the paper. However, the reviewer could not find any related discussion in the manuscript.

---
**The final update**

The paper initially received a rating of 6 because the reviewer, while expressing concerns about the reliability and reproducibility of the methodology as well as the rigor of the experimental section, appreciated the motivation behind the work, encapsulated in the statement: "not all videos are created equal." Based on this, the reviewer assigned a rating of 6.

The reasons for the two rating reductions

(1) During the rebuttal period, the author mentioned that modifying the setup from 100 timesteps to 30 timesteps led to a significant change in speedup, decreasing from 4.5x to 2.24x. This detail, however, was neither thoroughly discussed nor emphasized in the paper. It is evident that **the influence of this variable is much greater than that of resolution and video size, which were included in the ablation study**.
The absence of detailed discussion and emphasis on such experimental findings and conclusions not only risks confusing readers but also raises questions about the reliability and reproducibility of AdaCache. As a result, the reviewer lowered the rating for the first time.

(2) The lack of rigor in experimental comparisons

(i) The authors mentioned that the use of image conditions in OpenSora was motivated by an issue raised in the original repository, which stated that "the lack of a reference image leads to inconsistencies in video quality," i.e., a decline in motion consistency and visual quality. This implies that **the introduction of image conditions improves visual quality**.

(ii) The authors further claimed that AdaCache **directly utilized data from PAB because they reproduced PAB and obtained similar quantitative numbers** (with negligible changes). However, to the reviewer’s knowledge, PAB does not employ image conditions. **If the authors introduced image conditions to PAB for a fair comparison but still achieved similar VBench metrics, this appears highly counterintuitive.** If this is the case, what is the rationale behind AdaCache incorporating reference images? It should be noted that synthesizing reference images incurs significant computational overhead.

(iii) Given that a significant portion of the paper, including the experiments in the ablation study, was conducted on OpenSora, the inclusion of image conditions cannot be overlooked. The introduction of image conditions not only affects visual quality but may also impact the L1 distance between features, thereby influencing the caching rate. **This aspect should have been emphasized in the experimental section, yet it is not mentioned even once throughout the paper.**

(iv) To ensure fair quantitative comparisons, AdaCache should follow PAB's experimental setup, exclude reference images, and retest metrics such as VBench, SSIM, and PSNR.

Due to concerns about the rigor of the experiments, the reviewer downgraded the rating once again.

**Questions:**

1. The caching rate for the steps following step t is determined based on the rate of feature change between steps t and t+k. The reviewer wonders whether this metric is reasonable. For instance, as shown in Fig. 2, during the early and late stages of sampling, the L1 curve exhibits rapid changes, characterized by a large derivative. Would relying on the differences at earlier time steps to determine the subsequent caching rate introduce errors?
2. Could the authors provide more visual quality comparison results? For example, visualizations of the sampling process under different configurations (fast, slow) and how different video content leads to varying caching schedules, to more intuitively demonstrate the mechanism and effectiveness of the designed caching schedule.
3. PAB demonstrates impressive results in multi-GPU parallel processing. Can the authors' method leverage similar techniques (e.g., DSP) to scale to multi-GPU parallel inference? What would the efficiency be like?
4. The reviewer wants to know the source of the Delta-DIT performance in Table 1 and why there is almost no acceleration.

---

> ### Author Response · Authors · 2024-11-22
> **Response to reviewer g4jp [1/2]**
>
> **W1: Typo in Table 1**
>
> We sincerely apologize for this typo, and thank the reviewer for pointing it out. This SSIM value of AdaCache-slow in Open-Sora-Plan should be 0.7910 (instead of 79.10), consistent with other SSIM values. We will correct this in the final version of the paper.
>
>
> **W2: More details about caching-schedule hyperparameters.**
>
> We understand the reviewer’s concern and sincerely apologize for the lack of details. In AdaCache, once we compute the distance metric between subsequent representations ($c^l_t$), we select the next caching rate ($\tau^l_t$) based on a *pre-defined codebook of basis cache-rates*. Here, a *‘cache-rate’* is defined as the number of subsequent steps during which, a previously-computed representation is re-used (*i.e.,* a higher cache-rate gives more compute savings). Simply put, a higher distance metric will sample a lower cache-rate from the codebook, resulting in more-frequent re-computations.
>
> The codebook is basically a collection of cache-rates that is specific to a denoising schedule (i.e., #steps), coupled with distance metric ($c_t$) thresholds for selection. Both basis cache-rates and thresholds are hyperparameters. Here, optimal thresholds may need to be tuned per video-DiT baseline, whereas the cache-rates can be adjusted depending on the required speedup (*e.g.* AdaCache-fast, AdaCache-slow). We tune these hyperparameters (`codebook = {threshold-1: cache-rate-1, …}`) based on empirical observations on a small calibration set (with just 16 video prompts), and observe that they generalize well (*e.g.* on larger benchmarks such as VBench w/ 900+ prompts). This is thanks to the **normalized** cache-metric that we use for deciding the caching schedule (irrespective of the video prompt), relative to which we calibrate the threshold values.
>
> For instance, on Open-Sora baseline, we use the codebook `{0.03: 12, 0.05: 10, 0.07: 8, 0.09: 6, 0.11: 4, 1.00: 3}` in a 100-step denoising schedule, and the codebook `{0.08: 6, 0.16: 5, 0.24: 4, 0.32: 3, 0.40: 2, 1.00: 1}` for AdaCache-fast in a 30-step schedule. For AdaCache-slow in a 30-step schedule, we decrease the basis cache-rates (w/o having to change the thresholds), and use the codebook `{0.08: 3, 0.16: 2, 0.24: 1.00: 1}`. A specific cache-rate is selected if the distance metric is smaller than the corresponding threshold (and larger than any previous thresholds). We also ablate various codebooks (*e.g.* fast, mid, slow in Table 2e). We will include this discussion in the final version of the paper.
>
>
> **W3: Disparity between AdaCache vs. PAB comparisons in Table 1 and Fig 7.**
>
> We understand this perfectly-valid concern from the reviewer, let us clarify this confusion below.
>
> First, we want to point out that in Fig. 7, we compare AdaCache-fast (w/ MoReg) and PAB-fast configurations. In Table 1, if we consider these two configurations, we see that the quality metrics are not that different (*i.e.,* comparable), whereas AdaCache has much better speedups. AdaCache-slow is the variant that gives much better quality metrics, while still being faster than PAB-fast. Therefore, the quantitative numbers are consistent with the observations in Fig 7.
>
> However, we wish to highlight that a direct quality comparison based on Fig 7 is unfair, as AdaCache optimizes its latency to an extreme where the quality is expected to have a small drop. Yet, looking at Fig 5 we see that AdaCache performance is more-stable across a range of latencies, compared to PAB. A more reasonable setting would be to compare the quality at a similar latency, which we show in this [anonymous-fig-2](https://drive.google.com/file/d/1e30h_6N7K_QDcOHLRV0zCtNYqfnhlzuA/view?usp=share_link). Here, we include variants AdaCache (2.61x) vs. PAB (1.66x) for 720p - 2s generations, instead of a more-extreme variant AdaCache (4.49x) vs. PAB (1.26x) that we previously presented in Fig 7, making a more-fair comparison. We see that AdaCache shows a much better performance, still being faster.
>
> We will include this discussion and the figure for direct comparison in the final version of the paper. Also, with this rebuttal, we include an [anonymous-webpage](https://anonymous-adacache.github.io/), which we encourage reviewers to view. It includes many video comparisons, and provides a better view on baseline comparisons and ablations (*e.g.* how temporal consistency varies).

---

> > ### Author Response · Authors · 2024-11-25
> > **Follow-up**
> >
> > Dear Reviewer g4jp,
> >
> > Thank you again for your constructive feedback and time/effort reviewing our paper. Since the rebuttal period is ending soon, please let us know if our responses have addressed your concerns. We are happy to engage in further discussion to provide more clarifications if needed.
> >
> > Kind Regards!

---

> > > ### Comment · Reviewer_g4jp · 2024-11-26
> > >
> > > * The performance of AdaCache appears to be highly dependent on the codebook. It seems that the parameters of the codebook are manually configured. Is there any reliable and convincing guidance for determining an appropriate codebook? This might affect the usability and reproducibility of AdaCache.
> > > * The authors repeatedly mention that the codebook, calibrated using only 16 data samples, achieves good generalization. This is somewhat counterintuitive, as the paper demonstrates that different videos exhibit varying feature evolution processes. Are 16 samples sufficient to capture and cover all possible variation processes? What criteria should be used to select the calibration data?
> > > * The multi-GPU results appear somewhat unusual. According to the results reported by PAB, on OpenSora, four GPUs achieved over 6x speedup, and eight GPUs achieved over 10x speedup. However, the results you reported differ somewhat (e.g., only 3.96x speedup with four GPUs). Could you clarify this discrepancy?
> > > * The results for Delta-DiT and T-GATE in Table 1 are not missing. The reviewer is inquiring about the source of the unusual performance data.
> > > * Based on the scheduling process provided for the four samples, it appears that the caching frequency in the early stages of sampling is hardly restricted, and caching begins at a very early step.  Is this correct?  It is well-known that the early steps in the sampling process of diffusion models have a significant impact on the final results, where even minor perturbations can lead to substantial changes in the outcome.  For instance, during PAB’s broadcasting process, the early-stage feature sharing is deliberately avoided. **In this context, how is the performance of fidelity metrics (e.g., SSIM, PSNR) ensured to remain at a high level (better than PAB)?**

---

> > > > ### Author Response · Authors · 2024-11-27
> > > > **Follow-up response to reviewer g4jp [1/2]**
> > > >
> > > > We thank the reviewer for the engaged discussion, and allowing us resolve any further confusions. Please see our responses below.
> > > >
> > > > **F-Q2: Does the codebook calibrated on fewer video prompts generalize?**
> > > >
> > > > We observe this to be true in our experiments across multiple video-DiT baselines. In fact, in the table below, we show that the behavior of both quality metrics and speedups across different AdaCache variants is consistent in 32-video, 100-video and 900-video (standard VBench) benchmarks, validating that our hyperparameters within the codebook generalize well. Let us further clarify the reasoning for this.
> > > >
> > > > As mentioned before in our responses, the codebook consists of two sets of hyperparameters: (1) **basis cache-rates**, and (2) **cache-metric thresholds**.
> > > >
> > > > Among these, basis cache-rates can be set easily, depending on the speedup required by the user. We show that by simply changing the basis cache-rates (*w/o needing to tune the cache-metric thresholds*), AdaCache can achieve different speedups: fast, mid and slow (as given in Table 2e)— Here, cache-rates of AdaCache-mid (`8-6-4-2-1`) in row-2 of Table 2e correspond to the codebook `{0.03: 8, 0.05: 6, 0.07: 4, 0.09: 2, 0.11: 1, 1.00: 1}`, with *the same threshold values* as AdaCache-fast, and -slow. This shows that the proposed method supports a range of speedups w/o any further hyperparameter tuning.
> > > >
> > > > The cache-metric thresholds are the values we calibrate based on a small set of video prompts. We select these prompts randomly, and based on the distribution of metric values (*i.e.,* L1 distance between subsequent representations) across the denoising process, we select a reasonable range (*e.g.* 0.03 - 0.11 for 100-step Open-Sora) and split uniformly into the number of basis-rates we want to have. Since the cache-metric is **normalized**, the thresholds generalize well to unseen prompts. However, we agree that outliers could exist, yet on-average, we achieve a reasonable quality-latency trade-off as validated by our experiments.
> > > >
> > > > Such a generalization based on a small calibration set is not counterintuitive to our motivation that each video generation is unique (or, each video shows a unique variation in feature similarity— which also corresponds to the cache-metric). However, as validated in Fig 2-right, even though the distribution changes, the range of values (in y-axis) stay more-or-less the same. Meaning, the set of thresholds we calibrated can stay the same across different video generations, yet which threshold gets activated will vary depending on each video (based on the cache-metric). We will better clarify this in the supplementary.
> > > >
> > > > | Method | 32 videos || 100 videos || 900+ videos ||
> > > > |----------|----------|----------|----------|----------|----------|----------|
> > > > | 				| VBench| Latency (on A6000) |  VBench | Latency (on A6000) |  VBench | Latency (on A100) |
> > > > | Open-Sora			| **84.09** | **86.57** | **82.97** | **86.35** | 79.22 | 54.02 |
> > > > | + AdaCache-fast		| **83.42** | **37.06 (2.34x)** | **82.21** | **37.22 (2.32x)** | 79.39 | 24.16 (2.24x)  |
> > > > | + AdaCache-fast (w/ MoReg)	| **83.42** | **39.56 (2.19x)** | **82.32** | **39.65 (2.18x)** | 79.48 | 25.71 (2.10x)|
> > > > | + AdaCache-slow		| **83.93** | **57.33 (1.51x)** | **82.89** | **58.51 (1.48x)** | 79.66 | 37.01 (1.46x)|
> > > >
> > > > *all new numbers are in bold.*
> > > >
> > > >
> > > > **F-Q1: Guidance on deciding a codebook when adapting to a new setting (for usability and reproducibility).**
> > > >
> > > > As we discussed above, the basis cache-rates within our codebook are user-defined, and can be changed easily without any re-calibration depending on the required quality-latency trade-off. When setting the metric thresholds, we follow the simple strategy of (1) selecting a small calibration set of random video generation prompts, (2) observing the range of change in feature similarity, and (3) uniformly splitting such range into the set of basis cache-rates. A more-complex strategy (*e.g.* carefully sampling the calibration set, non-uniform splitting of range) may give better trade-offs with some extra effort. However, in our experiments, we observe that even a simpler strategy generalizes well to a large number of videos— thanks to (a) the consistent range of feature similarities between subsequent steps in a given DiT backbone, and (b) the *normalized* cache-metric. We will discuss these details in the supplementary. We also release our codebase with the paper to ensure the reproducibility and easier adoption to newer settings.

---

> > > > > ### Author Response · Authors · 2024-11-27
> > > > > **Follow-up response to reviewer g4jp [2/2]**
> > > > >
> > > > > **F-Q3: Disparity between multi-gpu results in [anonymous-fig-5] vs. those reported in PAB.**
> > > > >
> > > > > We sincerely apologize for the confusion here. The latency measurements (and, speedups) vary based on the video-generation settings (*e.g.* number of denoising steps, spatial resolution, number of frames) and the Hardware settings (*e.g.* type of GPU, memory availability, disk read/write speeds). It is important to run all baselines in the same setting (and, the same GPU node if possible) to make a fair comparison.
> > > > >
> > > > > For instance, in PAB [arXiv 2024], the authors run their multi-gpu experiments on 8xH100s, for generating 480p - 8s (204-frame) videos with Open-Sora using a 30-step schedule— showing a 10.5x speedup with 8 gpus. In accordance with our resource availability, we run our multi-gpu experiments on 8xA100s, for generating 480p - 2s (51-frame) videos with Open-Sora using a 30-step schedule— here, the same PAB baseline (based on the official codebase released by authors) shows a 5.24x speedup as given in [anonymous-fig-5](https://drive.google.com/file/d/1s7ECqxZ2NgZJuBcJCERbpUOVQxRgBnMg/view?usp=share_link). As long as we run AdaCache in the same setting, we ensure a fair comparison.
> > > > >
> > > > > Under such a fair comparison, we see that AdaCache consistently outperforms PAB in all gpu configurations. We will better clarify the experimental setting when we report these numbers in the final version of the paper.
> > > > >
> > > > >
> > > > > **F-Q4: Clarification on the performance of Delta-DiT / T-GATE in Table 1.**
> > > > >
> > > > > We sincerely apologize for our confusion about the reviewer’s original concern. Let us provide further information here. In Table 1, we reuse the baseline numbers reported by PAB [arXiv 2024], except for the latency measurements (and, speedups) which need to be replicated in our setting for a fair comparison— all the other metrics (VBench, PSNR, SSIM, LPIPS, FLOPs) are hardware independent and can be reused.
> > > > >
> > > > > The authors of PAB replicate Delta-DiT (from scratch) and T-GATE (from the official codebase), and adopt them to video generation, as detailed in the appendix of their paper. We believe that the limited speedups of these baselines are due to the fact that they are originally proposed as image-DiT acceleration methods, which do not fully exploit video information. Meaning, better speedups can not be achieved with these baselines without sacrificing the temporal consistency of generations. We verify that the numbers reported by PAB authors are correct, based on our T-GATE and PAB replications (both from the respective official codebases)— except for a reasonable variance in latency measurements which is expected. We will clarify where the numbers in Table 1 originate from, in the final version of the paper.
> > > > >
> > > > >
> > > > > **F-Q5: Why AdaCache does not have a hard boundary to avoid caching in early/late denoising steps?**
> > > > >
> > > > > In AdaCache, we do not have a hard boundary to avoid caching (either in very early/late denoising steps). As the reviewer stated, changes to the early denoising steps can change the convergence— and similarly, changes to the late denoising steps can introduce unwanted artifacts. However, since our method is adaptive, we do not have to avoid caching in these two-ends explicitly, but rather it is handled implicitly (as a soft boundary).
> > > > >
> > > > > For instance, in the above sample 3, the schedule is `[1, 2, 5, 8, 11, 14, 22, 30, 38, 46, 54, 62, 72, 82, 90, 98, 99, 100]`. In early steps (`[1, 2, 5, 8]`) the average cache-rate is 2.33. In late steps (`[ 90, 98, 99, 100]`), the average cache-rate is 3.33. In contrast, in middle steps (`[11, 14, 22, 30, 38, 46, 54, 62, 72, 82]`) the average cache-rate is 7.89. This shows that AdaCache will have a soft-boundary— that is also dependent on each video. We empirically observe that having such small cache-rates in early/late steps do not augment the convergence significantly, as also evident in our qualitative results.
> > > > >
> > > > > In PAB, the authors also observe the same behavior in feature similarity as in our Fig 2-right— early/late layers have higher change, whereas middle layers have lower change. Based on this, PAB introduce hard boundaries to avoid caching in early and late layers, and use a constant caching rate in the mid layers (as it is not adaptive)— which is a hand-designed schedule. In contrast, AdaCache provides more-flexibility (having adaptive soft-boundaries decided by the cache-metric), allowing us to optimize the latency even further, while preserving a better quality. We will better clarify the difference in the final version of the paper.

---

> > > > > > ### Comment · Reviewer_g4jp · 2024-11-27
> > > > > >
> > > > > > If the reviewer has correctly understood, AdaCache uses shared features at steps 3, 4, and 6, 7. Based on the reviewers’ experience, feature sharing in the early stages of sampling could often result in significant changes in the final synthesized content, which might be difficult to mitigate even with a lower caching rate. The paper mentions “reference-based” metrics such as PSNR and SSIM. Could the authors clarify whether any additional reference frames were introduced during inference to ensure high fidelity metrics in Open-Sora?

---

> ### Author Response · Authors · 2024-11-22
> **Response to reviewer g4jp [2/2]**
>
> **Q1: Is it reasonable to estimate the next cache-rate, based on previous features?**
>
> This is a very good question, and we thank the reviewer for raising it. We agree that relying on the rate-of-change computed based on past features, to adjust the subsequent cache schedule can introduce some errors (as the metric is lagged behind). However, this error is minimal. When observing the change between adjacent features as shown in Fig 2-right, we see that it varies smoothly, w/o any abrupt changes for the most-part of the denoising schedule. Therefore, relying on immediate-past features is not a bad idea.
>
> Regardless, if we want to avoid such errors, the solution that we can think of is having a dry-run. Meaning, we have to estimate the metrics in the first run, and then in the next run, adjust the caching-rate at each step based on an **up-to-date metric** (as we can not rely on future features to estimate an up-to-date metric on-the-fly). However, such an approach will beat the motivation of getting an inference acceleration. We will discuss this limitation in the supplementary.
>
>
> **Q2: Concrete examples of cache-schedules in different video generations.**
>
> We thank the reviewer for requesting this visualization, as it would provide better context to the reader. In this [anonymous-fig-4](https://drive.google.com/file/d/1g3bUI-g3tui4XvBSqwuW9x1iUoZfPB0U/view?usp=share_link), we show a few different video generations with their corresponding computational steps (within a 100-step baseline schedule), when accelerated with AdaCache-fast (w/ MoReg).
>
> Here, the first two videos have a smaller motion content, whereas the last two have higher motion. When observing the total number of compute steps, we see that it varies proportional to the motion content (*i.e.,* more motion → more compute steps). In terms of where the computations happen across the diffusion axis (*i.e.,* step-id), we see that every schedule is unique, supporting the underlying motivation of AdaCache. Here are the cache-shedules of these specific examples.
>
> Sample 1 (living-room): `15` compute-steps  @ step ids `[1, 2, 5, 8, 11, 14, 22, 32, 44, 56, 68, 80, 92, 99, 100]`
>
> Sample 2 (ancient-ruin): `14` compute-steps  @ step ids `[1, 2, 5, 8, 11, 17, 27, 39, 51, 63, 75, 87, 99, 100]`
>
> Sample 3 (moving-train): `18` compute-steps  @ step ids `[1, 2, 5, 8, 11, 14, 22, 30, 38, 46, 54, 62, 72, 82, 90, 98, 99, 100]`
>
> Sample 4 (underwater): `16` compute-steps  @ step ids `[1, 2, 5, 8, 11, 17, 27, 35, 45, 55, 65, 75, 85, 95, 99, 100]`
>
> We will include and discuss these observations in the final version of the paper.
>
>
> **Q3: AdaCache performance on multi-gpu settings.**
>
> We thank the reviewer for raising this interesting comparison. The benefit of AdaCache is especially relevant in multi-gpu settings, as it not only reduces computational costs, but also avoids some of the *gpu communication overheads*. In this rebuttal, we evaluate the acceleration on multiple-gpus and report in this [anonymous-fig-5](https://drive.google.com/file/d/1s7ECqxZ2NgZJuBcJCERbpUOVQxRgBnMg/view?usp=share_link). Here, we reily on Dynamic Sequence Parallelism (DSP), and compare Open-Sora and Open-Sora-Plan baselines, with eather PAB or AdaCache for acceleration. Here, AdaCache consistently outperforms PAB with better inference speeds across all settings. We will include these results in the final version of the paper.
>
>
> **Q4: Clarification on missing latency values (Delta-DiT, T-GATE) in Table 1.**
>
> We sincerely apologize for the confusion here. It is not the case that Delta-DiT and T-GATE have no acceleration, but rather the latency values could not be replicated for these baselines in our settings (at least at the time of submission). To be more specific, Delta-DiT has no publicly-available codebase, which prevents us from replicating and measuring its latency. However, in this rebuttal, we provide new latency measurements for T-GATE as it provides an open-source implementation: it costs 49.11s (1.10x speedup) w/ Open-Sora, 113.75s (1.14x speedup) w/ Open-Sora-Plan, and w/ Latte 29.23s (1.11x speedup). We will update these results in Table 1.

---

> ### Author Response · Authors · 2024-11-27
> **Follow-up response (2) to reviewer g4jp**
>
> We really appreciate the engaged discussions from the reviewer, and we are happy to porvide further clarifications.
>
> **F2-Q1: Which features are shared among which-steps?**
>
> We apologize for any confusion here. We believe the reviewers understanding is correct here. In the above example, we note the *'compute-steps'* to be `[1, 2, 5, 8, 11, ...]`. This means AdaCache reuses the *residual features* computed in step `2`, through steps `3,4`, whereas new residual features are computed in step `5` to be reused in through subsequent steps `6,7`. We wish to highlight that only the residual computations (as shown in Fig 4-right) are reused, whereas the iteratively-denoised representation gets updated in every step (either based on recomputed or cached+reused residual features).
>
> We uderstand the reviewers concern on diverging from the original baseline generation. This behavior is noticiable in some qualitative examples that we provide (*e.g.* in Fig 7 - bottom row, Fig 6 - middle row), yet not significantly affecting quantitative numbers (in Table 1). We further encourage the revierwer to visit our [anonymous-webpage](https://anonymous-adacache.github.io/), that includes many video results--- inclduing diverging cases (*e.g.* bottom of the webpage, middle column). We hope we provide sufficient evidence to address this concern.
>
> **F2-Q2: Clarification on reference-based metrics.**
>
> We sincerely apologize for the confusion here. We introduce some metrics (*e.g.* PSNR, SSIM, LPIPS) as *'reference-based'*, as they are computed relative to a baseline--- in this case, relative to the corresponding DiT baseline w/o any acceleration. These metrics are computed the same way for all acceleration methods that we report (*e.g.* Delta-DiT, T-GATE, PAB), where we do nothing different for AdaCache.
>
> In terms of our experimental setup, we follow the exact same settings as the original baselines (w/ the only exception being the introduction of the proposed caching schedule), in all our video generation pipelines--- Open-Sora, Open-Sora-Plan and Latte. In other words, we do not introduce any unfair advantage in AdaCache to preserve the generation quality when optimizing the latency. We hope this clarifies the reviewer's concern.
>
> Please let us know if further information is needed.

---

> > ### Author Response · Authors · 2024-12-02
> > **Follow-up**
> >
> > We thank the reviewer g4jp again for their continued enagement in the discussions. Please let us know if the concerns have been addressed. Since the rebuttal period is ending soon, we would really appreciate it if our additional experiments and clarifications can be considered in the final rating.
> >
> > Thanks so much!

---

> > > ### Author Response · Authors · 2024-12-03
> > > **Follow-up (2)**
> > >
> > > Dear Reviewer g4jp,
> > >
> > > We thank you again for your initial feedback and continued engagement in discussions. Please let us know if your concerns have been addressed. Since the rebuttal period is ending soon, we would really appreciate it if our additional experiments and clarifications can be considered in the final rating.
> > >
> > > Thanks so much!

---

> > > > ### Author Response · Authors · 2024-12-04
> > > > **Final response to reviewer g4jp**
> > > >
> > > > We thank the reviewer for the engaged discussion. Yet, we find it really unfortunate that the reviewer decides to reduce the rating even further (6 -> 5 -> 3), even after the rebuttal period has ended, not allowing the authors to engage in further discussions. We have tried our best to accommodate all the reviewer requests including extensive experimentation and detailed clarifications throughout the two-week discussion period, spending a significant effort.
> > > >
> > > > Let us provide our final responses to the reviewer below.
> > > >
> > > >
> > > > **Q 1(i): The reliability of quantitative comparisons.**
> > > >
> > > > In our main results (Table 1), we first run the PAB baseline on VBench benchmark and verify that we can replicate the same quantitative numbers (w/ a negligible change)--- as we find no negligible change, we report the same numbers as in the PAB paper for the quality metrics, and recompute all the latency measurements on our hardware (as latency changes depending on hardware). Then only we adopt the same baseline settings to run AdaCache. We believe we make a fair and reliable comparison across all the baselines.
> > > >
> > > > **Q 1(ii): Delta-DiT and T-GATE numbers in Table 1 are unusual.**
> > > >
> > > > As mentioned in our previous responses, we adopt the same numbers for these two image-DiT baselines as reported in the PAB paper. We do not re-implement them ourselves. We direct the reviewer to appendix A.3 in PAB paper for a detailed description on the re-implementation settings. We believe that the limited speedups of these baselines are due to the fact that they are proposed as image-DiT acceleration methods— which do not fully exploit video information to be competitive in video-DiT acceleration.
> > > >
> > > > **Q 2(i): Clear discussion on T2V and TI2V settings.**
> > > >
> > > > We experiment across these two families of models (T2V and TI2V) to show the generalization of AdaCache. In the rebuttal, we provide evidence of further generalization (w/ T2I and multi-modal T2V settings). We will include this discussion on different families of models in the final version of the paper.
> > > >
> > > > **Q 2(ii) Analysis of different speedups at different configurations (*e.g.* 4.5x at 100-steps, 2.24x at 30-steps in Open-Sora)**
> > > >
> > > > Throughout the paper, when we report speedups, we are always upfront about the corresponding video generation configurations— we mention the spatial resolution, frame count and the number denoising steps. We believe without these details, the acceleration measurements are not grounded. By experimenting on different configurations, we provide the reader a holistic-view on how much speedup to be expected.
> > > >
> > > > As discussed in our previous responses, the speedup of AdaCache depends on different settings such as (1) number of denoising steps, (2) resolution/frame-count in the generation, and (3) the architecture of the underlying DiT. We believe the above-mentioned variation is not unusual for an adaptive, training-free acceleration method that optimizes quality-latency trade-off. We will include this discussion in the final version of the paper.
> > > >
> > > > **Based on all our responses during this two-week discussion period, we kindly request the reviewer to reconsider their rating. We believe we have addressed most of the reviewer concerns, and the reject rating (6 -> 3) is unreasonable.**

---

### Official Review · Reviewer_5tUr · 2024-10-27

**Soundness:** 4
**Presentation:** 4
**Contribution:** 4
**Rating:** 5
**Confidence:** 5

**Summary:**

The paper introduces Adaptive Caching (AdaCache), a plug-and-play, training-free method to accelerate video generation using diffusion transformers.  The authors note that "not all videos are created equal", meaning some videos don’t need as many processing steps to reach high quality. Based on this, they propose an adaptive caching strategy that reduces the computation of denoising steps based on the rate-of-change. The authors also introduce a Motion Regularization (MoReg) scheme to adjust the caching schedule, which can allocate computation based on video motion content for improving the quality-latency trade-off. Experimental results demonstrate that AdaCache significantly speeds up existing video diffusion models.

**Strengths:**

a. The approach presented is straightforward, and the method section is generally clear and easy to follow.
b. The motivation for the work is reasonable and interesting, i.e., "not all videos are created equal".
c. AdaCache provides a training-free acceleration method that can be applied to existing video diffusion models, achieving significant speedups without additional model training.

**Weaknesses:**

1. Lines 285-287 mention that using unique caching schedules for each layer makes the generations unstable, but it’s unclear why this is the case. It would help if the authors provided an explanation.
2. Equation 5 introduces a codebook for the caching rate, but it’s not clear what this codebook is or how it’s created. The authors should add more details to clarify this part of the method.
3. While Table 1 shows AdaCache outperforming PAB, the qualitative comparison in Fig. 7 shows a different result. AdaCache seems to lose more visual detail, especially in the details. This raises concerns about its practical quality compared to PAB.
4. In Table 1, AdaCache achieves better VBench results than the baseline. The authors should explain why the accelerated video have better results, especially since the visual quality in Fig. 7 is noticeably worse than the baseline.
5.  In Table 1, the SSIM of AdaCache-slow on Line 346 appears unusually high.

I’m concerned about the fairness of the experiments. On the OpenSora model, PAB results are based on text-to-video task, while AdaCache is tested on image-to-video task.  I keep my original Rating.

**Questions:**

1. The primary concern with this paper is the practical effectiveness of AdaCache. More qualitative comparisons are needed to robustly demonstrate AdaCache’s effectiveness in preserving visual quality, especially for the detail generations.
2. The authors need to provide more detailed methodological details, such as the construction and role of the codebook in caching rates.

---

> ### Author Response · Authors · 2024-11-22
> **Response to reviewer 5tUr [1/2]**
>
> **W1: Why per-layer caching-schedules can be unstable?**
>
> We thank the reviewer for requesting further evidence on this observation, which we believe will be useful to the reader. By design, AdaCache can have unique caching schedules *per-layer* (and, per each residual computation). However, we observe that it will make the generations unstable in the current DiT architectures that we tested.
>
> One possible hypothesis for this observation is the incompatibility between the cached and newly-computed features. Having unique caching schedules *per-layer*, forces the model to use both these features within **different DiT layers of the same denoising step**. As such features (cached vs. recomputed) are not perfectly-aligned, their compatibility becomes an issue which results in an unstable performance.
>
> In contrast, if we have a common caching schedule for all the layers, the features used in each step (either cached or recomputed) would all correspond to **a specific same denoising step**, that are all perfectly-compatible with each other. We include qualitative samples in this rebuttal to visualize this observation (see [anonymous-fig-1](https://drive.google.com/file/d/1a_VzVZh82hFE-5ZN1HIA6vrIA7WoHiX_/view?usp=share_link)). We see that the per-layer cache-schedule shows more artifacts/degradations in the generated videos, compared to a more-stable common cache-schedule.
>
> However, we note that this observation may not generalize to all video generation models (*e.g.* some architectures may be stable enough to take advantage of unique levels of redundancy in different layers), so we keep formulation/design of AdaCache more-generic in the method section. We will include this discussion in the supplementary.
>
>
> **W2: More details about caching-schedule hyperparameters.**
>
> We understand the reviewer’s concern and sincerely apologize for the lack of details. In AdaCache, once we compute the distance metric between subsequent representations ($c^l_t$), we select the next caching rate ($\tau^l_t$) based on a *pre-defined codebook of basis cache-rates*. Here, a *‘cache-rate’* is defined as the number of subsequent steps during which, a previously-computed representation is re-used (*i.e.,* a higher cache-rate gives more compute savings). Simply put, a higher distance metric will sample a lower cache-rate from the codebook, resulting in more-frequent re-computations.
>
> The codebook is basically a collection of cache-rates that is specific to a denoising schedule (i.e., #steps), coupled with distance metric ($c_t$) thresholds for selection. Both basis cache-rates and thresholds are hyperparameters. Here, optimal thresholds may need to be tuned per video-DiT baseline, whereas the cache-rates can be adjusted depending on the required speedup (*e.g.* AdaCache-fast, AdaCache-slow). We tune these hyperparameters (`codebook = {threshold-1: cache-rate-1, …}`) based on empirical observations on a small calibration set (with just 16 video prompts), and observe that they generalize well (*e.g.* on larger benchmarks such as VBench w/ 900+ prompts). This is thanks to the **normalized** cache-metric that we use for deciding the caching schedule (irrespective of the video prompt), relative to which we calibrate the threshold values.
>
> For instance, on Open-Sora baseline, we use the codebook `{0.03: 12, 0.05: 10, 0.07: 8, 0.09: 6, 0.11: 4, 1.00: 3}` in a 100-step denoising schedule, and the codebook `{0.08: 6, 0.16: 5, 0.24: 4, 0.32: 3, 0.40: 2, 1.00: 1}` for AdaCache-fast in a 30-step schedule. For AdaCache-slow in a 30-step schedule, we decrease the basis cache-rates (w/o having to change the thresholds), and use the codebook `{0.08: 3, 0.16: 2, 0.24: 1.00: 1}`. A specific cache-rate is selected if the distance metric is smaller than the corresponding threshold (and larger than any previous thresholds). We also ablate various codebooks (*e.g.* fast, mid, slow in Table 2e). We will include this discussion in the final version of the paper.

---

> ### Author Response · Authors · 2024-11-22
> **Response to reviewer 5tUr [2/2]**
>
> **W3: Disparity between AdaCache vs. PAB comparisons in Table 1 and Fig 7.**
>
> We understand this perfectly-valid concern from the reviewer, let us clarify this confusion below.
>
> First, we want to point out that in Fig. 7, we compare AdaCache-fast (w/ MoReg) and PAB-fast configurations. In Table 1, if we consider these two configurations, we see that the quality metrics are not that different (*i.e.,* comparable), whereas AdaCache has much better speedups. AdaCache-slow is the variant that gives much better quality metrics, while still being faster than PAB-fast. Therefore, the quantitative numbers are consistent with the observations in Fig 7.
>
> However, we wish to highlight that a direct quality comparison based on Fig 7 is unfair, as AdaCache optimizes its latency to an extreme where the quality is expected to have a small drop. Yet, looking at Fig 5 we see that AdaCache performance is more-stable across a range of latencies, compared to PAB. A more reasonable setting would be to compare the quality at a similar latency, which we show in this [anonymous-fig-2](https://drive.google.com/file/d/1e30h_6N7K_QDcOHLRV0zCtNYqfnhlzuA/view?usp=share_link). Here, we include variants AdaCache (2.61x) vs. PAB (1.66x) for 720p - 2s generations, instead of a more-extreme variant AdaCache (4.49x) vs. PAB (1.26x) that we previously presented in Fig 7, making a more-fair comparison. We see that AdaCache shows a much better performance, still being faster.
>
> We will include this discussion and the figure for direct comparison in the final version of the paper. Also, with this rebuttal, we include an [anonymous-webpage](https://anonymous-adacache.github.io/), which we encourage reviewers to view. It includes many video comparisons, and provides a better view on baseline comparisons and ablations (*e.g.* how temporal consistency varies).
>
>
> **W4: Why AdaCache archives better quantitative numbers than the baseline in Table 1, while there are noticeable artifacts in Fig 7?**
>
> We understand this valid concern. First, we would like to highlight that the results better than the baseline in Table 1 are achieved by AdaCache-slow, whereas the visualizations that we present in Fig 7 are with AdaCache-fast (w/ MoReg)— that optimizes latency to an extreme, where the quality is expected to show a small drop. Please see this [anonymous-fig-2](https://drive.google.com/file/d/1e30h_6N7K_QDcOHLRV0zCtNYqfnhlzuA/view?usp=share_link) for a fair comparison between AdaCache-slow and Baseline for 720p - 2s generations. We will better clarify this in the paper.
>
> In case of AdaCache-slow, we hypothesize such better quantitative numbers are shown due to two potential reasons: (1) by reusing representations across multiple steps, the denoising process gets smoothed-out, resulting in fewer sharp changes in noise predictions— which we have observed to be an issue with Open-Sora baseline. (2) the quantitative metrics do not perfectly-align with the perceived visual quality— which has been observed in many prior work (and the reason why such work also evaluate models based on human preferences).
>
> To alleviate the above issue (2), in this rebuttal, we also conduct a user preference study to measure the quality of adacache generations, comparing AdaCache against prior-work and the baseline. Here, we collect a total of 1800 responses from 36 different users in the form of A/B preference tests. The results of this study is given in this [anonymous-fig-3](https://drive.google.com/file/d/1G7L5-KHTk3Yf76cGYldHERawmgN5L49l/view?usp=share_link). Between AdaCache and PAB, we see a clear win for our method (70%) while being extremely-similar to the baseline more than half the time (41%). Among AdaCache variants, users find these to beoften tied (60%) in-terms of perceived quality, yet still showing a better preference for motion-regularized variant (25% vs. 14%). This study validates the effectiveness of Adaptive Caching, and shows that it is indistinguishable from the baseline in many examples. We will include this study and discussion in the final version of the paper.
>
>
> **W5: Typo in Table 1**
>
> We sincerely apologize for this typo, and thank the reviewer for pointing it out. This SSIM value of AdaCache-slow in Open-Sora-Plan should be 0.7910 (instead of 79.10), consistent with other SSIM values. We will correct this in the final version of the paper.

---

> ### Author Response · Authors · 2024-11-25
> **Follow-up**
>
> Dear Reviewer 5tUr,
>
> Thank you again for your constructive feedback and time/effort reviewing our paper. Since the rebuttal period is ending soon, please let us know if our responses have addressed your concerns. We are happy to engage in further discussion to provide more clarifications if needed.
>
> Kind Regards!

---

> > ### Author Response · Authors · 2024-11-28
> > **Follow-up (2)**
> >
> > We thank the reviewer 5tUr again for the initial feedback, which will definitely improve the quality and clarity of this paper. We believe we have addressed all of the reviewer's concerns, but we would be very happy to engage in further discussion and provide more clarifications if needed. Please let us know if the concerns have been addressed.
> >
> > Thanks so much!

---

> > > ### Comment · Reviewer_5tUr · 2024-12-02
> > > **Official Comment by Reviewer 5tUr**
> > >
> > > Thanks for the response!
> > > I have some concerns: In the publicly available code, the authors utilized a pre-inferred initial image as the reference image input for OpenSora. However, under the experimental setup without using a reference image, the 100-step inference on OpenSora did not achieve the expected performance in terms of acceleration and visual quality. Could the authors clarify the cause of this discrepancy?

---

> ### Author Response · Authors · 2024-12-02
> **Follow-up response to reviewer 5tUr**
>
> We thank the reviewer for the engaged discussion, and allowing us resolve any further confusions.
>
> We sincerely apologize for the confusion here. In our AdaCache experiments, we follow the baseline video-DiT inference setup exactly (except for the caching-related changes), in all Open-Sora, Open-Sora-Plan, and Latte pipelines. Among these, Open-Sora is both text- and image-conditioned, as suggested by the original contributors in their github issues (please see [issue-1](https://github.com/hpcaitech/Open-Sora/issues/504) and [issue-2](https://github.com/hpcaitech/Open-Sora/issues/550)) and GradIO demo (please see [instructions-to-run-locally](https://github.com/hpcaitech/Open-Sora/tree/main/gradio) as the public demo is currently offline). In contrast, in Open-Sora-Plan and Latte, the video generations are only text-conditioned. By being faithful to each setting, we show that AdaCache can generalize to both these settings. We release our reference implementation publicly with this paper, including the detailed steps to replicate the reported results, validating our contributions based on the Open-Sora baseline. We will continue to update our codebase to support other video-DiT baselines that we experimented with.
>
> We thank the reviewer for the time and effort spent on these discussions. We hope that we were able to address all the concerns, and the reviewer will kindly consider this fact in the final rating.

---

> > ### Author Response · Authors · 2024-12-02
> > **Follow-up**
> >
> > We thank the reviewer 5tUr again for their continued enagement in the discussions. Please let us know if the concerns have been addressed. Since the rebuttal period is ending soon, we would really appreciate it if our additional experiments and clarifications can be considered in the final rating.
> >
> > Thanks so much!

---

> > > ### Author Response · Authors · 2024-12-03
> > > **Follow-up (2)**
> > >
> > > Dear Reviewer 5tUr,
> > >
> > > We thank you again for your initial feedback and continued engagement in discussions. Please let us know if your concerns have been addressed. Since the rebuttal period is ending soon, we would really appreciate it if our additional experiments and clarifications can be considered in the final rating.
> > >
> > > Thanks so much!

---

> > > > ### Comment · Reviewer_5tUr · 2024-12-03
> > > > **Official Comment by Reviewer 5tUr**
> > > >
> > > > Thank you for the response.
> > > >
> > > > In the publicly available code, during the 100-step inference of OpenSora in the OpenSora gallery, when image condition is not used, the inference efficiency (Speedup) and visual fidelity (SSIM, PSNR; not VBench) did not meet expectations. (1) Could the authors clarify whether this discrepancy is as expected? If so, what is the source of this difference (aside from the performance differences of the model itself in T2V and I2V tasks)? (2) PAB works well without the image condition (the default configuration of PAB does not include the image condition). Could AdaCache achieve stable and consistent performance without image condition? If so, why is an initial image needed before inference? (3) In Table 1, are the OpenSora results based on both image and text conditions? Why is the SSIM of OpenSora significantly higher than that of Open-Sora-Plan and Latte? (4) In the ablation study, are the OpenSora results based on both image and text conditions?

---

> ### Author Response · Authors · 2024-12-03
> **Follow-up response to reviewer 5tUr**
>
> We thank the reviewer for the continued enagement in discussions. Let us answer the reviewer questions below.
>
> > In the publicly available code, during the 100-step inference of OpenSora in the OpenSora gallery, when image condition is not used, the inference efficiency (Speedup) and visual fidelity (SSIM, PSNR; not VBench) did not meet expectations. (1) Could the authors clarify whether this discrepancy is as expected? If so, what is the source of this difference (aside from the performance differences of the model itself in T2V and I2V tasks)?
>
> This should not be the expected behavior. AdaCache preserves a better quality-latency trade-off compared to other inference acceleration pipelines, regardless of whether image-conditioned or not (as also validated by our experiments on different model variants in Table 1).
>
> We are unsure which configuration the reviewer experimented with. However, we want to highlight our Fig. 5, where we show that in extreme cases (AdaCache-fast), reference-based metrics (*e.g* SSIM, PSNR) are expected to drop. Still, we show much better trade-offs compared to PAB. Moreover, our qualitative results validate that the perceived visual quality is preserved even in such extreme cases, revealing the limitations of such reference-based metrics.
>
> > (2) PAB works well without the image condition (the default configuration of PAB does not include the image condition). Could AdaCache achieve stable and consistent performance without image condition? If so, why is an initial image needed before inference?
>
> In all our experimental settings, we compare with PAB, and show that AdaCache consistently achieves much better quality-latency trade-offs (please also see our [anonymous-webpage](https://anonymous-adacache.github.io/) for video results). This is regardless of being image-conditioned or not. We also want to highlight that a direct comparison of visual quality should be made at similar speedups (as shown in webpage above), where AdaCache achieves superior performance.
>
> As mentioned previously, we stay faithful to the original inference setting of each video-DiT baseline. In Open-Sora this corresponds to both image- and text-conditioned generation (as mentioned by its original contributors in their GradIO demo and github issues [issue-1](https://github.com/hpcaitech/Open-Sora/issues/504) and [issue-2](https://github.com/hpcaitech/Open-Sora/issues/550)). In Open-Sora-Plan and Latte, this corresponds to just text-conditioned generation. By experimenting on both these settings, we validate that AdaCache generalizes and achieves a stable performance in both settings.
>
> > (3) In Table 1, are the OpenSora results based on both image and text conditions? Why is the SSIM of OpenSora significantly higher than that of Open-Sora-Plan and Latte?
>
> We follow the original inference settings suggested by the original contributors— in OpenSora, this is image- and text-conditioned generation.
>
> The original quality metrics depend on how good each baseline is. In our experiments, we observe that Open-Sora gives much-better generations than other baselines, which results in better quality metrics. This behavior is also observed in the results reported in PAB paper (and other concurrent work such as FasterCache).
>
> > (4) In the ablation study, are the OpenSora results based on both image and text conditions?
>
> As also mentioned above, we follow the original inference settings suggested by the original contributors— in OpenSora, this is image- and text-conditioned generation.
>
> **We thank the reviewer for the time and effort spent on these discussions. We hope that we were able to address all the concerns, and the reviewer will kindly consider this fact in the final rating.**

---

### Official Review · Reviewer_KT6w · 2024-11-01

**Soundness:** 4
**Presentation:** 4
**Contribution:** 3
**Rating:** 8
**Confidence:** 5

**Summary:**

The paper introduces a training-free method called Adaptive Caching (AdaCache) to accelerate video Diffusion Transformers (DiTs). AdaCache is based on the idea that "not all videos are created equal," meaning some videos require fewer denoising steps to achieve reasonable quality. It caches computations through the diffusion process and devises a caching schedule tailored to each video generation to maximize the quality-latency trade-off. Additionally, the paper introduces a Motion Regularization (MoReg) scheme to utilize video information within AdaCache, essentially controlling compute allocation based on motion content. These plug-and-play contributions significantly speed up inference (e.g., up to 4.7× faster on Open-Sora 720p - 2s video generation) without compromising generation quality across multiple video DiT baselines. The code for this method will be made publicly available.

**Strengths:**

1. Adaptive Caching achieve very good performance even compared with recent PAB paper. I very appreciate it.

2. This approach requires no training and can seamlessly be integrated into a baseline video DiT at inference, as a plug-and-play component.

3. Motion Regularization (MoReg) to allocate computations based on the motion content in the video being generated seems to be very reasonable.

**Weaknesses:**

1. Regarding the choice of metric, why was the Mean Squared Error (MSE) selected directly? Can the MSE metric truly reflect the actual reduction in features between adjacent steps? Are there alternative metrics that might be more suitable, or can you provide comparisons with other metrics such as the cosine similarity metric or others?

2. Secondly, I'm interested in knowing if the proposed method is compatible with large Text-to-Image (T2I) base models, like FLUX. If it is, what would be the expected impact on the performance metrics?

3. Although above questions exists, I think this is a really valuable paper

**Questions:**

On the whole, I consider this paper to be well-executed. However, I'm intrigued by the possibility of identifying a metric that could more accurately assess the redundancy in the system. Furthermore, I'm curious about the potential of integrating layer-wise broadcasting dynamically with distillation techniques to enhance real-time video generation capabilities.

---

> ### Author Response · Authors · 2024-11-22
> **Response to reviewer KT6w [1/1]**
>
> **W1: Is L1/L2 distance, the right choice to compute the cache-metric?**
>
> We understand this valid concern. In AdaCache, we want to measure the rate-of-change in computed features (*i.e.,* residual connections in STA, CA, MLP layers) across the diffusion steps, so that we can make a decision on when to cache-and-reuse/recompute features. If the change is small— meaning the features are highly-redundant— then we can reuse previously cached features in subsequent steps.
>
> To measure the difference between such features, we need to rely on a distance metric that (1) is fast to compute, and (2) measures an absolute distance which directly corresponds to the given input (*i.e.,* we can not rely on distribution-based distances such as KL-divergence). Among fast and direct distance measures (*e.g.* L1, L2, Cosine-distance), we see that L1/L2 give an absolute measure which aligns better with the actual change. In contrast, Cosine-distance computes a normalized-distance, which is not a reasonable estimate of change. For instance, if the features differ only by a scale, the cosine distance will be zero. However, here we wish to have a non-zero value as the features have actually changed. We ablate these different metrics in Table 2c of the original paper (and discussed in L471-476), which verify the better performance of absolute distance metrics such as L1/L2. Among these, L1 provides a better quality-latency trade-off, and hence, we adopt it by default. We will include this extended discussion in the supplementary.
>
>
> **W2: AdaCache performance with Image-DiTs (e.g. DiT-XL/2 or PixArt-alpha).**
>
> We agree that this comparison based on image-DiTs is useful to evaluate how AdaCache generalizes to image generation pipelines. We believe AdaCache will still provide reasonable speedups, but we expect the acceleration to be smaller than that of a video-DiT, which relies of heavier operators (*e.g.* spatial-temporal attention) within the baseline.
>
> To validate this, we are currently experimenting with the DiT-XL/2 baseline, and will report the results in the subsequent comments within the coming days. We really appreciate the patience of the reviewer as the experiments are being finalized.

---

> ### Author Response · Authors · 2024-11-26
> **Follow-up: AdaCache on an image-DiT baseline**
>
> **W2: AdaCache performance with Image-DiTs (e.g. DiT-XL/2).**
>
> Following the reviewer’s suggestion, in this rebuttal, we implement AdaCache (w/o Motion Regularization) on top of an image generation baseline: DiT-XL/2. In the table below, we observe that AdaCache gives reasonable speedups. As expected, the acceleration is smaller than that with a video-DiT in similar settings, which rely on heavier operators (*e.g.* spatial-temporal attention) within the baseline. This shows that AdaCache (originally proposed for accelerating video generation) can also generalize to image generation pipelines.
>
> | Method 	| FID $\downarrow$ 	| sFID $\downarrow$ 	| IS $\uparrow$ 	| Precision $\uparrow$ 	| Recall $\uparrow$  | Latency (s) $\downarrow$ 	| Speedup $\uparrow$ 	|
> |----------|----------|----------|----------|----------|----------|----------|----------|
> | DIT-XL/2 	| 2.30	| 4.56	| 276.56	| 0.83		| 0.58	| **16.15** 	| **1.00x**	|
> | + AdaCache	| **3.27**	| **7.19**	| **243.21**| **0.79**		| 0.59	| **5.98**	| **2.70x**	|
>
> *all new numbers are in bold.*

---

> > ### Comment · Reviewer_KT6w · 2024-11-29
> > **To authors**
> >
> > Your responses have solved my concern. I keep my original score.

---

> > > ### Author Response · Authors · 2024-11-29
> > > **Thanks to reviewer KT6w**
> > >
> > > We are happy that we were able to resolve all the concerns of reviewer KT6w. We thank the reviewer again for the time/effort spent reviewing our paper, and the positive rating.

---

### Official Review · Reviewer_vGd3 · 2024-11-03

**Soundness:** 2
**Presentation:** 2
**Contribution:** 2
**Rating:** 6
**Confidence:** 4

**Summary:**

The paper proposes training-free Diffusion Transformer (DiT) acceleration named AdaCache. The method is motivated by the fact that different videos require different amounts of computation. AdaCache decides whether to skip or recompute the cache during the cache step defined by the schedule, using an introduced rate-of-change metric $c_t$, which is basically L1 distance between residual block features in current and previous cache schedule timesteps.  Authors further augment rate-of-change metric c_t with a Motion Regularization (MoReg) metric to add information about the motion content of the generated video.

Overall, the AdaCache idea has good potential. However, the current version of the manuscript needs significant revision to be accepted for this conference. Please refer to the Weaknesses and Questions sections for more details.

**Strengths:**

1. Novelty: The idea of adaptive caching seems novel in the field of diffusion model caching.
2. Motivation: The paper provides a clear motivation for AdaCache method.
3. Clearness: The method is simple and easy to understand.

**Weaknesses:**

1. Method section requires clarifications:

a. The paper lacks information about the selection of rate-of-change schedule hyperparameters.

b. Lines 286-287 stat that authors observe that unique caching schedules for each layer will make the generations unstable. This important observation requires further explanation and clarification.

2. Experiment results require better presentation:

a. There are concerns regarding the reported speedup and latency. Given that AdaCache is not a deterministic method and inference time for different videos varies, it is incorrect to report just the mean inference time and speedup for all videos. Standard deviation (std) values should be included.

b. Ablation studies are performed on only 32 videos. Considering that AdaCache is not deterministic, the results may have low statistical significance.

c. AdaCache without MoReg is a general method that could apply to image diffusion transformers. A comparison with prior works on DiT architectures for image generation, such as PIXART-α for T2I generation and DiT-XL for class-conditional image generation on ImageNet, is suggested.

d. The paper missed a comparison with another DiT caching method like FORA [1].

3. The paper lacks Limitation section. It would be interesting to see if there are videos on which AdaCache performs worse than its deterministic competitors.

**Questions:**

1. My main concerns regarding this paper are related to experiments results presentation and clarification of method hyperparameters:

a. Since method is not deterministic, the results should include std values for inference time and speedup (See Weakness 2a). Moreover, apart of adding std values, I recommend to conduct ablation studies on more than 32 videos (See Weakness 2b).

b. AdaCache without MoReg is not tied to video generation, I recommend to include image generation DiTs caching (See Weakness 2c).

c. I suggest authors to provide information about rate-of-change schedule hyperparameters selection, as it is crucial pert of the proposed method (See Weakness 1a).

2. Additional group of questions is not as important as the main one, but can also help to improve the quality of the manuscript:

a. Statement regarding unique caching schedules for each layer needs clarification (See Weakness 1a).

b. The paper would greatly benefit from method’s limitations analysis (See Weakness 2).

c. The authors may include FORA [1] in comparisons.

d. It interesting to see how AdaCache performs on MMDiT models such as CogVideoX [2] and SD-3 [3].

e. In line 196, it would be beneficial to explain which features were used for visualization.


[1] Selvaraju, P., Ding, T., Chen, T., Zharkov, I., & Liang, L. (2024). Fora: Fast-forward caching in diffusion transformer acceleration. arXiv preprint arXiv:2407.01425.

[2] Yang, Z., Teng, J., Zheng, W., Ding, M., Huang, S., Xu, J., ... & Tang, J. (2024). Cogvideox: Text-to-video diffusion models with an expert transformer. arXiv preprint arXiv:2408.06072.

[3] Esser, P., Kulal, S., Blattmann, A., Entezari, R., Müller, J., Saini, H., ... & Rombach, R. (2024, March). Scaling rectified flow transformers for high-resolution image synthesis. In Forty-first International Conference on Machine Learning.

---

> ### Author Response · Authors · 2024-11-22
> **Response to reviewer vGd3 [1/3]**
>
> **W1a: More details about caching-schedule hyperparameters.**
>
> We understand the reviewer’s concern and sincerely apologize for the lack of details. In AdaCache, once we compute the distance metric between subsequent representations ($c^l_t$), we select the next caching rate ($\tau^l_t$) based on a *pre-defined codebook of basis cache-rates*. Here, a *‘cache-rate’* is defined as the number of subsequent steps during which, a previously-computed representation is re-used (*i.e.,* a higher cache-rate gives more compute savings). Simply put, a higher distance metric will sample a lower cache-rate from the codebook, resulting in more-frequent re-computations.
>
> The codebook is basically a collection of cache-rates that is specific to a denoising schedule (i.e., #steps), coupled with distance metric ($c_t$) thresholds for selection. Both basis cache-rates and thresholds are hyperparameters. Here, optimal thresholds may need to be tuned per video-DiT baseline, whereas the cache-rates can be adjusted depending on the required speedup (*e.g.* AdaCache-fast, AdaCache-slow). We tune these hyperparameters (`codebook = {threshold-1: cache-rate-1, …}`) based on empirical observations on a small calibration set (with just 16 video prompts), and observe that they generalize well (*e.g.* on larger benchmarks such as VBench w/ 900+ prompts). This is thanks to the **normalized** cache-metric that we use for deciding the caching schedule (irrespective of the video prompt), relative to which we calibrate the threshold values.
>
> For instance, on Open-Sora baseline, we use the codebook `{0.03: 12, 0.05: 10, 0.07: 8, 0.09: 6, 0.11: 4, 1.00: 3}` in a 100-step denoising schedule, and the codebook `{0.08: 6, 0.16: 5, 0.24: 4, 0.32: 3, 0.40: 2, 1.00: 1}` for AdaCache-fast in a 30-step schedule. For AdaCache-slow in a 30-step schedule, we decrease the basis cache-rates (w/o having to change the thresholds), and use the codebook `{0.08: 3, 0.16: 2, 0.24: 1.00: 1}`. A specific cache-rate is selected if the distance metric is smaller than the corresponding threshold (and larger than any previous thresholds). We also ablate various codebooks (*e.g.* fast, mid, slow in Table 2e). We will include this discussion in the final version of the paper.
>
>
>
> **W1b: Why per-layer caching-schedules can be unstable?**
>
> We thank the reviewer for requesting further evidence on this observation, which we believe will be useful to the reader. By design, AdaCache can have unique caching schedules *per-layer* (and, per each residual computation). However, we observe that it will make the generations unstable in the current DiT architectures that we tested.
>
> One possible hypothesis for this observation is the incompatibility between the cached and newly-computed features. Having unique caching schedules *per-layer*, forces the model to use both these features within **different DiT layers of the same denoising step**. As such features (cached vs. recomputed) are not perfectly-aligned, their compatibility becomes an issue which results in an unstable performance.
>
> In contrast, if we have a common caching schedule for all the layers, the features used in each step (either cached or recomputed) would all correspond to **a specific same denoising step**, that are all perfectly-compatible with each other. We include qualitative samples in this rebuttal to visualize this observation (see [anonymous-fig-1](https://drive.google.com/file/d/1a_VzVZh82hFE-5ZN1HIA6vrIA7WoHiX_/view?usp=share_link)). We see that the per-layer cache-schedule shows more artifacts/degradations in the generated videos, compared to a more-stable common cache-schedule.
>
> However, we note that this observation may not generalize to all video generation models (*e.g.* some architectures may be stable enough to take advantage of unique levels of redundancy in different layers), so we keep formulation/design of AdaCache more-generic in the method section. We will include this discussion in the supplementary.

---

> ### Author Response · Authors · 2024-11-22
> **Response to reviewer vGd3 [2/3]**
>
> **W2a: As AdaCahe is not deterministic, report standard deviation of latency measurements.**
>
> We thank the reviewer for raising this valid concern. In AdaCache, the variation in latency is small. Hence, following the standard practice as in other adaptive methods (*e.g.* AdaDiff, Object-Centric Diffusion, LazyDiffusion, Block Caching), we initially reported the average latency on a standard benchmark. However, we agree that the standard deviation (*std*) would provide useful information to the reader. Therefore, in this rebuttal, we include *std* numbers for AdaCache with the Open-Sora baseline. We will include them in our results tables in the final version of the paper.
>
> | Method | Latency (s) |
> |----------|----------|
> | Open-Sora	| 54.02	|
> | + AdaCache-fast	| 24.16 $\pm$ **1.54** |
> | + AdaCache-fast (w/ MoReg)	| 25.71  $\pm$ **1.08**	|
> | + AdaCache-slow	| 37.01  $\pm$ **1.30**	|
>
> *all new numbers are in bold.*
>
>
> **W2b: Does ablations on 32 videos generalize?**
>
> We understand the reviewer’s point of view. In this paper we consider standard benchmark video prompts in all evaluations (VBench prompts for 900+ videos in Table 1 and Open-Sora-Gallery prompts for 32 videos in Table 2), keeping all the experiments reproducible. We decide to have a smaller benchmark for ablations, to keep the overall computational cost tractable as we evaluate a wide range of design decisions. However, we find that the observations usually generalize across both benchmarks.
>
> To further strengthen this claim, we provide additional results on a new set of prompts (corresponding to 100 videos), comparing 480p-2s video generations. We include both mean and standard deviation values for AdaCache variants as their latency is dependent on each video generation. Here, we make two observations:
>
> (1) Even though the absolute performance (VBench) metrics vary between benchmarks— which is expected as the set of prompts are different for each setting— the overall change between different model variants stays consistent: AdaCache-slow performs better than AdaCache-fast, and MoReg helps improve the performance.
>
> (2) The standard deviation in latency measurements is small, in all benchmarks. This shows that the speedups that we report in the ablation table generalize to the larger benchmarks such as VBench (900+ videos).
>
> **Note:** we have to rely on A6000 gpus for newly-reported latency measurements as we no longer have access to original the A100 gpus, yet the speedups remain consistent.
>
> We will include these results and the discussion in the final version of the paper.
>
> | Method | 32 videos || 100 videos || 900+ videos ||
> |----------|----------|----------|----------|----------|----------|----------|
> | | VBench | Latency (on A6000) |  VBench | Latency (on A6000) |  VBench | Latency (on A100) |
> | Open-Sora	| **84.09** | **86.57** | **82.97** | **86.35** | 79.22 | 54.02 |
> | + AdaCache-fast	| **83.42** | **37.06 $\pm$ 0.89** | **82.21** | **37.22 $\pm$ 0.70** | 79.39 | 24.16 **$\pm$ 1.54**  |
> | + AdaCache-fast (w/ MoReg)	| **83.42** | **39.56 $\pm$ 0.94** | **82.32** | **39.65 $\pm$ 1.16** | 79.48 | 25.71 **$\pm$ 1.08** |
> | + AdaCache-slow	| **83.93** | **57.33 $\pm$ 1.53** | **82.89** | **58.51 $\pm$ 1.61** | 79.66 | 37.01 **$\pm$ 1.30** |
>
> *all new numbers are in bold.*

---

> ### Author Response · Authors · 2024-11-22
> **Response to reviewer vGd3 [3/3]**
>
> **W2c: AdaCache performance with Image-DiTs (e.g. DiT-XL/2 or PixArt-alpha).**
>
> We agree that this comparison based on image-DiTs is useful to evaluate how AdaCache generalizes to image generation pipelines. We are currently experimenting with DiT-XL/2 baseline, and will report the results in the subsequent comments within the coming days. We really appreciate the patience of the reviewer as the experiments are being finalized.
>
>
>
> **W2d: Missing citation/comparison with image-DiT acceleration method: FORA.**
>
> We thank the reviewer for bringing this important related work to our attention. First, we kindly note that FORA (released on arXiv in July 2024) is considered a “non-penalized concurrent work” as per the ICLR 2025 submission policy. That being said, we still believe this would be a valuable discussion to include when reporting the performance of AdaCache on image-DiT baselines (as we are currently experimenting with DiT-XL/2). Conceptually FORA is different from AdaCache, as it is a caching mechanism proposed purely for image-DiTs, and is not adaptive w.r.t. the input.
>
> We will add a quantitative comparison with FORA in the subsequent comments within the coming days. We really appreciate the patience of the reviewer as the experiments are being finalized.
>
>
> **Q1d: AdaCache performance with Multi-modal DiTs (e.g. CogVideoX).**
>
> We agree that this comparison based on multi-modal DiTs is useful to evaluate how AdaCache generalizes to various DiT pipelines. We are currently experimenting with CogVideoX baseline, and will report the results in the subsequent comments within the coming days. We really appreciate the patience of the reviewer as the experiments are being finalized.
>
>
> **W3: Discussion on the limitations of AdaCache (e.g. worse performing settings compared to prior work).**
>
> This is an important discussion which we will include in the final version of the paper.
>
> First, we note that AdaCache usually outperforms the quality of other inference optimization methods at a comparable speedup, as validated by the quantitative (*e.g.* Fig 5) and many qualitative results that are already in the paper. In this rebuttal, we also include an [anonymous-webpage](https://anonymous-adacache.github.io/), which we encourage reviewers to view as it includes many video comparisons to support this claim (better-viewed on Chrome browser). However, when we further reduce the latency— considerably beyond that of the prior work— we start seeing some artifacts and loss of fine-grained detail (*e.g.* as visible in some examples in Fig 7). Yet, we highlight that a fair comparison should be ideally made at comparable speedups (see [anonymous-fig-2](https://drive.google.com/file/d/1e30h_6N7K_QDcOHLRV0zCtNYqfnhlzuA/view?usp=share_link))
>
> In addition, we observe a few limitations of the current AdaCache implementation:
>
> (1) As we do not rely on any re-training (or, finetuning) of the baseline model (which gives considerable compute savings and data acquisition costs), any limitations that are present in the corresponding baseline may transfer to the AdaCache variant of the same model. It is important that we raise caution about this to the user.
>
> (2) In the current setup, the hyperparameters related to the caching schedule (*e.g.* basis cache-rates, cache metric thresholds) are set based on heuristics and empirical validation on a small set of video prompts. Although these generalize well as we observe in our experiments, they may require some tuning when adopting to different baseline models or denoising schedules.
>
> (3) Finally, as our computational graph is adaptive, it may be less-suited in custom hardware architectures that rely on fixed (*i.e.,* static) computational graphs for running model inference (*e.g.* custom chips for on-device inference). AdaCache variant with a fixed caching schedule (tuned with a pre-defined calibration dataset) will work better in such scenarios.
>
> **Q1e: Which features used for visualization in Fig 2 (right)?**
>
> We sincerely apologize for the lack of details about the features that we use in Fig 2-right (and L196). Here, we consider residual computations corresponding to *spatial-temporal attention* within an Open-Sora baseline for 720p - 2s video generations. We select a pre-defined layer (here, the middle-layer of the DiT), sample the features, and compute L1-distance between corresponding features of subsequent diffusion steps (aggregated over all axes) to come up with a scaler representation of feature change. Based on how this metric varies, we can get an idea how redundant these computations are during different stages of denoising. We will better-clarify and include these details in the final version of the paper.

---

> ### Author Response · Authors · 2024-11-26
> **Follow-up: AdaCache on an image-DiT baseline**
>
> **W2c, W2d: AdaCache performance with Image-DiTs (e.g. DiT-XL/2) and comparison with FORA [arXiv, July 2024].**
>
> Following the reviewer’s suggestion, in this rebuttal, we implement AdaCache (w/o Motion Regularization) on top of an image generation baseline: DiT-XL/2, and compare with the concurrent work FORA [arXiv, July 2024]. Conceptually, FORA is different from AdaCache, as it is a caching mechanism proposed purely for accelerating image-DiTs (not extended to video generation), and is not adaptive w.r.t. the input. In the table below, we observe that AdaCache shows a better/comparable performance with FORA on all quantitative metrics. Please see the trade-off curve in [anonymous-fig-6](https://drive.google.com/file/d/17SXQmYtw7ufPdrwJfIoI8vjA-MOjQKU-/view?usp=share_link), that shows how AdaCache outperforms FORA at the same latency. This shows that AdaCache (originally proposed for accelerating video generation) can also generalize to image generation pipelines.
>
> | Method 	| FID $\downarrow$ 	| sFID $\downarrow$ 	| IS $\uparrow$ 	| Precision $\uparrow$ 	| Recall $\uparrow$  | Latency (s) $\downarrow$ 	| Speedup $\uparrow$ 	|
> |----------|----------|----------|----------|----------|----------|----------|----------|
> | DIT-XL/2 	| 2.30	| 4.56	| 276.56	| 0.83		| 0.58	| **16.15** 	| **1.00x**	|
> | + FORA (Thres=3)	| 2.82	| 6.04	| 253.96	| 0.80		| 0.58	| **6.68**	| **2.42x**	|
> | + FORA (Thres=5)	| 4.97	| 9.15	| 222.97| 0.76		| 0.59	| **4.80**	| **3.36x**	|
> | + AdaCache	| **3.27**	| **7.19**	| **243.21**| **0.79**		| 0.59	| **5.98**	| **2.70x**	|
>
> *all new numbers are in bold.*

---

> ### Author Response · Authors · 2024-11-28
> **Follow-up: AdaCache on a multi-modal DiT baseline**
>
> **Q1d: AdaCache performance with Multi-modal DiTs (e.g. CogVideoX).**
>
> Following the reviewer’s suggestion, in this rebuttal, we implement AdaCache on top of a milti-modal diffusion transformer for video generation: CogVideoX, and compare with the concurrent work FasterCache [arXiv, Oct 2024]— a training-free inference acceleration method that is not content adaptive. Here, we generate 480p - 6s videos following the baseline, and evaluate on prompts from Open-Sora gallery. In the table below, we observe that AdaCache shows a better quality-latency trade-off compared to FasterCache, and validate that it can work with multi-modal DiTs.
>
> | Method | VBench | Latency (s) | Speedup |
> |----------|----------|----------|----------|
> | CogVideoX-2B				| **82.20**	|**152.70**		| **1.00x**	|
> | + FasterCache [arXiv, Oct 2024]	| **82.13**	|**102.32**		| **1.49x**	|
> | + AdaCache-fast			| **82.00**	|**92.51**		| **1.65x**	|
> | + AdaCache-slow			| **82.46**	|**102.47**		| **1.49x**	|
>
> *all new numbers are in bold.*

---

> > ### Comment · Reviewer_g4jp · 2024-11-29
> >
> > On OpenSora, with the same prompt, AdaCache-fast achieves a 4.7× speedup. Why does it only achieve 1.65× on CogVideoX? Could the authors clarify the reasons for this significant discrepancy?
> >
> > Furthermore, AdaCache achieves a speedup of 4.7× on prompts from the OpenSora gallery, yet only 2.24× on prompts from VBench. What accounts for such a significant discrepancy in the speedup achieved by the same model?

---

> ### Author Response · Authors · 2024-11-28
> **Follow-up**
>
> We believe, now we have addressed all of the reviewer's concerns, but we would be very happy to engage in further discussion and provide more clarifications if needed.

---

> > ### Comment · Reviewer_vGd3 · 2024-11-29
> >
> > Dear Authors,
> >
> > Thank you for conducting extensive experiments and providing clarifications to my questions. I still have several points that need clarification to form a detailed opinion about your paper:
> >
> > 1. It is still unclear to me how to select a codebook of basis cache rates for new diffusion architectures. Could you please provide an algorithm or detailed steps on how to achieve this?
> >
> > 2. The generalization from small to large numbers of videos remains ambiguous. You mentioned using a "small calibration set (with just 16 video prompts)" to select codebook parameters, which seems to contradict your paper's motivation that “not all videos are created equal.” If the calibration set mainly includes complex videos, the selected codebook might not be aggressive enough for simpler videos that could be cached more efficiently. How do you ensure the calibration set is representative?
> >
> > Looking forward to your clarifications.
> >
> > Best regards,
> > Reviewer vGd3

---

> ### Author Response · Authors · 2024-11-29
> **Follow-up response to reviewer g4jp**
>
> **F1-Q1: Disparity between the speedups in Open-Sora (Table 2b, Table 1) and CogVideoX experiments.**
>
> We really appreciate the engaged discussions from the reviewer g4jp, and we are happy to porvide further clarifications.
>
> We sincerely apologize for the confusion, let us clarify here. The speedups that we observe depend on factors such as the size of generated videos (*e.g.* spatial resolution, number of frames), the denoising schedule, and the undelying DiT model achitecture (+ scale).
>
> For instance, in Table 2b, we see a 4.7x speedup for **720p - 2s (51-frame)** video generations with a baseline denoising schedule of **100** steps. This is the standard generation setup followed by Open-Sora baseline contributors. However, In Table 1, we follow the same experimental setup introduced in PAB [arXiv 2024] to ensure a fair comparison. In this setting, for Open-Sora baseline, we generte **480p - 2s (51-frame)** videos with a baseline denoising schedule with **30** steps (showing 2.24x speedup). We already clarify the details of each setting in the corresponding table captions, and will better highlight how these will affect the speedups in the final version of the paper.
>
> In the new experiments with CogVideoX-2B, we follow the correspoding original setup of generating **480p - 6s (49-frame)** videos with a baseline denoising schedule of **50** steps (showing 1.65x speedup). We will include all such details when we report these results in the paper.
>
> Please let us know if further clarifications are required.

---

> ### Comment · Reviewer_g4jp · 2024-11-29
>
> The authors’ clarification regarding the impact of video resolution and number of frames on the speedup factor is not convincing. For example, in Table 2(b) of the main text, consistent and stable speedup factors are observed across different resolutions and frame rates. The significant speedup variation caused solely by different timestep settings raises concerns about the reliability and stability of AdaCache. As a result, the reviewer has decided to lower the rating.

---

> ### Author Response · Authors · 2024-11-29
> **Follow-up response (2) to reviewer g4jp**
>
> We thank the reviewer for the engaged discussion. Let us clarify further.
>
> In the comment above, we were discussing all different factors affecting the speedup (even as relatively-small changes). As shown in Table 2b, the spatial resolution and number of frames do incur a relatively small variation, giving a stable performance (4.5x in 480p - 2s setting, 4.4x in 480p - 4s setting and 4.7x in 720p - 2s setting).
>
> Yet, the original denoising schedule have a more impact on the speedup. For instance, in a 100-step schedule, the rate-of-change between subsequent features is smooth/small. Hence, AdaCache can afford to reuse representations for a longer period (giving higher speedups). In contrast, in a 30-step schedule, the rate-of-change is relatively-larger, and if the same representations are re-used for a longer period, it incurs considerable quality degradations. Therefore, we decide our AdaCache setting such that we get the best quality-latency trade-off in each setting.
>
> We also highlight that in all settings, we outperform prior similar acceleration methods--- both in-terms of quality and latency, across multiple benchmarks and baseline DiTs--- validating the generalizability of AdaCache.
>
> We kindly ask the reviewer to reconsider the decision to lower their rating in light of this discussion. We have tried our best to answer all reviewer concerns during the rebuttal period with significant efforts (including additional experimental settings and evaluations). Please give us an opportunity to to provide further clarifications as needed.

---

> ### Author Response · Authors · 2024-11-30
> **Follow-up response to reviewer vGd3**
>
> We really appreciate the engaged discussions from the reviewer vGd3, and we are happy to provide further clarifications.
>
> **F-Q1: How to select the codebook hyperparameters when adapting to a new setting?**
>
> Our codebook consists of two sets of hyperparameters: (a) **cache-metric thresholds**, and (b) **basis cache-rates**. We follow the steps below when tuning these:
>
> (1) Select a small calibration set of random video generation prompts. We select 16 prompts and visually validate the varying levels of complexity in the corresponding video generations (*e.g.* high- and low-frequency textures, fast and slow moving content)— We use the same set of prompts across all our experimental settings, observing that these generalize. We will highlight our calibration set in the paper for easier adaptability.
>
> (2) Observe the distribution of cache-metric values across the denoising process (*i.e.,* L1 distance between subsequent representations), and identify the lower- and upper-bounds— We identify `[0.03, 0.11]` to be this range for the 100-step Open-Sora baseline.
>
> (3) Split the above range uniformly into the number of *basis cache-rates* we want to have— We use the *cache-metric thresholds* `{0.03, 0.05, 0.07, 0.09, 0.11}` for the 100-step Open-Sora baseline.
>
> (4) Finally, set the *basis cache-rates* depending on the required quality-latency trade-off— These values are user-defined and can be adjusted at inference without needing to tune anything else. We use *basis cache-rates* `12-10-8-6-4-3` for AdaCache-fast with the 100-step Open-Sora baseline (*i.e.,* the codebook will be `{0.03: 12, 0.05: 10, 0.07: 8, 0.09: 6, 0.11: 4, 1.00: 3}`). For AdaCache-mid in Table 2e, we use basis rates `8-6-4-2-1-1` with the same thresholds (*i.e.,* the codebook will be `{0.03: 8, 0.05: 6, 0.07: 4, 0.09: 2, 0.11: 1, 1.00: 1}`). Using smaller basis rates will yield a better quality, but also a smaller speedup. These basis rates can be adjusted based on the number of denoising steps, and validated by inspecting the change in quality of the generated videos (either visually or quantitatively).
>
> We use the same process (w/ the same prompts) across Open-Sora, Open-Sora-Plan, Latte and CogVideoX baselines, achieving consistently-better quality-latency trade-offs compared to prior training-free DiT acceleration methods. We will better highlight these steps in the paper. We also release our codebase with the paper to ensure the reproducibility and easier adoption to newer settings.
>
>
> **F-Q2: On the generalization of the codebook tuned with fewer video prompts.**
>
> We observe that our codebook generalizes across multiple benchmarks. In the table below, we show that both the quality metrics and speedups in different AdaCache variants behave consistently across 32-video, 100-video and 900-video (standard VBench) benchmarks--- all using the codebook tuned with the same 16 prompts. Let us further clarify the reasoning for this.
>
> | Method | 32 videos || 100 videos || 900+ videos ||
> |----------|----------|----------|----------|----------|----------|----------|
> | 				| VBench| Latency (on A6000) |  VBench | Latency (on A6000) |  VBench | Latency (on A100) |
> | Open-Sora			| 84.09 | 86.57 | 82.97 | 86.35 | 79.22 | 54.02 |
> | + AdaCache-fast		| 83.42 | 37.06 (2.34x) | 82.21 | 37.22 (2.32x) | 79.39 | 24.16 (2.24x)  |
> | + AdaCache-fast (w/ MoReg)	| 83.42 | 39.56 (2.19x) | 82.32 | 39.65 (2.18x) | 79.48 | 25.71 (2.10x)|
> | + AdaCache-slow		| 83.93 | 57.33 (1.51x) | 82.89 | 58.51 (1.48x) | 79.66 | 37.01 (1.46x)|
>
>
> First, we ensure a fair spread of calibration prompts by visually validating the corresponding video generations to have varying levels of complexity (*e.g.* high- or low-frequency textures, fast and slow moving content). Secondly, by making our cache-metric a **normalized** one, we make sure that the thresholds that we tune generalize well to unseen prompts (for a given DiT model, and denoising schedule).
>
> This not counterintuitive to our motivation that each video generation is unique. Even though the range of normalized metric values stays the same across different video generations, the underlying distribution of values is still **unique for each video** (*e.g.* how the metric changes in different stages of denoising)— as also seen in Fig 2-right. Meaning, the thresholds we calibrated can stay the same across different video generations, yet which one gets activated at a given step will vary depending on each video. We will better clarify this in the supplementary.
>
> That being said, we agree with the reviewer that outliers could exist, and bettter speedups may be squeezed with a more-complex codebook selection. Yet on-average, we achieve a quality-latency trade-off that consistently outperforms prior acceleration methods as validated by our experiments.
>
> Please let us know if further clarifications are required, as we are happy to engage in further discussions. We thank the reviewer again for the time and effort spent on these discussions.

---

> ### Comment · Reviewer_vGd3 · 2024-11-30
>
> Dear Authors,
>
> Thank your prompt response to by questions. However, some moments of your explanation need more formal definitions and detailed explanations:
>
> **F-Q1: Selection of Codebook Hyperparameters in New Settings**
>
> 1. The step "Observe the distribution of cache-metric values across the denoising process" is unclear. The L1 distance between subsequent representations can vary across architectures. Please provide a more generalized definition for selecting lower and upper bounds.
>
> 2. How should users define the number of basis cache-rates? What is its impact, and can you recommend values for this parameter?
>
> 3. Basis cache-rates involve numerous hyperparameters, making optimization complex. Could you provide detailed instructions to simplify this process?
>
> **F-Q2: Generalization of the Codebook Tuned with Fewer Video Prompts**
>
> 1. How do you define "a fair spread of calibration prompts"? This is crucial for the method’s applicability.
>
> 2. In the provided table, how were the subsets of 32 and 100 videos selected?
>
>
> Best regards,
> Reviewer vGd3

---

> ### Author Response · Authors · 2024-12-02
> **Follow-up response (2) to reviewer vGd3 [1/2]**
>
> We really appreciate the engaged discussions from the reviewer vGd3, and we are happy to provide further clarifications.
>
>
> **F-Q1: Selection of Codebook Hyperparameters in New Settings**
> >  The step "Observe the distribution of cache-metric values across the denoising process" is unclear. The L1 distance between subsequent representations can vary across architectures. Please provide a more generalized definition for selecting lower and upper bounds.
>
> We sincerely apologize for the lack of clarity here. By “observing the distribution”, we mean visualizing the histograms of cache-metric across the denoising process (as shown in Fig. 2-right). These histograms (when averaged across the calibration set) provide the information such as lower- and upper-bound of the cache-metric. When adapting to a new setting (*e.g.* DiT architecture, or denoising schedule), we simply rely on such histograms to set our range of thresholds in the codebook. We will include the visualization script (and a script that outputs the range) with the release of our codebase, outlining clear steps for adapting the range of cache-metrics to newer settings.
>
>
> > How should users define the number of basis cache-rates? What is its impact, and can you recommend values for this parameter?
>
> In our experiments, we decide the number of basis cache-rates heuristically: we find that, having 2-6 basis cache-rates works well in practice, depending on the *granularity of caching* we need.
>
> For instance, when caching to an extreme (*e.g.* AdaCache-fast, where we cache up to 12-steps at a time, in a 100-step schedule), there is a higher-chance of getting severe artifacts in the accelerated model. Hence, having more fine-grained thresholds (*i.e.,* higher number of basis cache-rates), helps us control the number of cached steps and avoid such artifacts. In contrast, in settings with minimal acceleration (*e.g.* AdaCache-slow where we cache only up to 2-steps), we can have fewer number of basis cache-rates.
>
> Our recommendation is to decide the number of cache-rates based on the required acceleration. With a higher acceleration, having fine-grained basis cache-rates helps better preserve the quality. The required acceleration can be decided based on the quality-latency constraints of the user application, and AdaCache has the flexibility to support such varying configurations. We will discuss this guideline in the final version of the paper.
>
>
> > Basis cache-rates involve numerous hyperparameters, making optimization complex. Could you provide detailed instructions to simplify this process?
>
> We agree with the reviewer’s concern, and let us provide our high-level approach for defining basis cache-rates.
>
> (1) Decide the required level of acceleration (*e.g.* fast, slow or mid) based on the quality-latency requirements of user application.
>
> (2) Set the largest basis-cache rate based on the above, and select 2-6 rates to split the range of basis cache-rates depending on the required caching granularity (*e.g.* finer granularity is better with higher acceleration).
>
> (3) Run the accelerated model on the calibration set, and evaluate the quality-latency metrics (*e.g.* VBench quality, and wall-clock time).
>
> (4) Adjust the basis cache-rates heuristically, and iterate above (2)-(3) steps.
>
> *e.g.* (a) if the required acceleration is not met, increase the largest basis-cache rate,
>
> *e.g.* (b) if the required quality-level is not met, increase the caching granularity—*i.e.,* the number of basis cache-rates— or decrease the largest basis-cache rate.
>
> This iterative process of hyperparameter tuning is relatively-faster, as we rely on a small calibration set and an accelerated inference pipeline. We can even parallelize this process as a common grid search. We will detail these steps in the final version of the paper.

---

> ### Author Response · Authors · 2024-12-02
> **Follow-up response (2) to reviewer vGd3 [2/2]**
>
> **F-Q2: Generalization of the Codebook Tuned with Fewer Video Prompts**
> > How do you define "a fair spread of calibration prompts"? This is crucial for the method’s applicability.
>
> We thank the reviewer for raising this question. As we discuss in our motivation (Section 3), we observe that the complexity of video generations can be characterized across spatial and temporal variations in generated content. In terms of the spatial axis, we select prompts that result in either homogeneous textures, or high-frequency textures. In terms of the temporal axis, we select prompts that result in either a small or a large motion content. By including these 4 types of videos in our calibration set, we construct a fair spread in terms of video generation complexity (which in-turn, results in a range of optimal compute requirements). We will include a complete and separate discussion about the calibration set in the supplementary, and release the corresponding prompts with our codebase.
> > In the provided table, how were the subsets of 32 and 100 videos selected?
>
> Here, the set of 32 videos comes from the standard Open-Sora gallery prompts (given [here](https://hpcaitech.github.io/Open-Sora/)) and the set of 100 videos comes from publicly-available Sora prompts (given [here](https://promptsora.com/)). In contrast, the set of 900+ videos correspond to standard VBench prompts (given [here](https://github.com/Vchitect/VBench/blob/master/prompts/all_dimension.txt)). We will clarify these subsets when reporting these numbers in the final version of the paper
>
>
> We thank the reviewer for the time and effort spent on these discussions. We hope that we were able to address the reviewer's concerns. We also hope that our additional experiments and clarifications will be kindly considered in the final rating.

---

> > ### Author Response · Authors · 2024-12-02
> > **Follow-up**
> >
> > We thank the reviewer vGd3 again for their continued enagement in the discussions. Please let us know if the concerns have been addressed. Since the rebuttal period is ending soon, we would really appreciate it if our additional experiments and clarifications can be considered in the final rating.
> >
> > Thanks so much!

---

> > > ### Comment · Reviewer_vGd3 · 2024-12-02
> > >
> > > Dear Authors,
> > >
> > > Thank you for your efforts during the discussion period. I found your idea of adaptive caching innovative and the results interesting, leading me to raise my rating to 6.
> > >
> > > However, the manuscript needs some revisions. Specifically, your explanations clarified the choice of hyperparameters, which are crucial for reproducibility and should be included in the revised paper. Additionally, the complexity of these hyperparameters may pose a challenge for first-time users of AdaCache, suggesting a promising direction for future research. I also recommend incorporating the experimental results shared during the rebuttal period into the manuscript.
> > >
> > > Best regards,
> > > Reviewer vGd3

---

> > > > ### Author Response · Authors · 2024-12-02
> > > > **Thanks to reviewer vGd3**
> > > >
> > > > We are happy that we were able to resolve the concerns of reviewer vGd3. We thank the reviewer again for the time and effort spent reviewing our paper and engaging in lengthy discussions, allowing us to provide much-needed clarifications. These discussions will imporve the quality of our paper greatly, which we will include in the final version of the paper. Finally, we appreciate the reviewer's positive rating.
> > > >
> > > > Kind Regards!

---

### Author Response · Authors · 2024-11-22
**General comment**

We thank all the reviewers for their constructive feedback and appreciate their time/effort reviewing our paper. In this rebuttal, we provide clarifications with evidence to answer reviewer concerns, as individual responses to each reviewer. Please let us know if further clarifications are needed during the rebuttal period.

---

### Meta-Review · Area_Chair_E4UD · 2024-12-15

**Metareview:**

The paper introduces a novel adaptive algorithm but has several weaknesses. The choice of MSE as a metric is questioned, and the reviewer asks whether alternative metrics, like cosine similarity, would be more suitable. The compatibility with large T2I models like FLUX is unclear. The methodology section lacks clarity, especially regarding the pre-defined codebook. The qualitative comparison is insufficient, with AdaCache showing worse performance on certain datasets, raising concerns about stability. The reliability of the quantitative comparison is questionable, particularly regarding the experimental setup and data sources in Table 1. The experimental details, such as conditions for Open-Sora-Plan and Latte, should be clarified, and the speedup factor variation needs further discussion.

The review scores are mixed, but the detailed negative feedback from the reviewers highlights significant issues with the paper. After reviewing the paper and rebuttal, the Area Chair also concluded that the paper should be rejected.

**Additional Comments On Reviewer Discussion:**

The reviewers appreciate the motivation behind the work but expressing concerns about the reliability and rigor of the methodology and experiments. The remained concerns are significant: 1) The significant impact of changing the timesteps from 100 to 30 (speedup dropping from 4.5x to 2.24x) was not adequately discussed or emphasized in the paper, raising concerns about the reliability and reproducibility of AdaCache. 2) The experimental comparisons lacked rigor.

---

### Decision · Program_Chairs · 2025-01-22

Reject